# Continuous sensing of nutrients and growth factors by the mTORC1-TFEB axis

**Breanne Sparta‡, Nont Kosaisawe†, Michael Pargett†, Madhura Patankar, Nicholaus DeCuzzi, John G Albeck***

Department of Molecular and Cellular Biology, University of California, Davis, Davis, United States

**Abstract** mTORC1 senses nutrients and growth factors and phosphorylates downstream targets, including the transcription factor TFEB, to coordinate metabolic supply and demand. These functions position mTORC1 as a central controller of cellular homeostasis, but the behavior of this system in individual cells has not been well characterized. Here, we provide measurements necessary to refine quantitative models for mTORC1 as a metabolic controller. We developed a series of fluorescent protein-TFEB fusions and a multiplexed immunofluorescence approach to investigate how combinations of stimuli jointly regulate mTORC1 signaling at the single-cell level. Live imaging of individual MCF10A cells confirmed that mTORC1-TFEB signaling responds continuously to individual, sequential, or simultaneous treatment with amino acids and the growth factor insulin. Under physiologically relevant concentrations of amino acids, we observe correlated fluctuations in TFEB, AMPK, and AKT signaling that indicate continuous activity adjustments to nutrient availability. Using partial least squares regression modeling, we show that these continuous gradations are connected to protein synthesis rate via a distributed network of mTORC1 effectors, providing quantitative support for the qualitative model of mTORC1 as a homeostatic controller and clarifying its functional behavior within individual cells.

**\*For correspondence:**
jgalbeck@ucdavis.edu

†These authors contributed equally to this work

**Present address:** ‡Department of Integrative Biology and Physiology, University of California, Los Angeles, Los Angeles, United States

## Editor's evaluation

In this study, Sparta et al., performed time-resolved monitoring of the mechanistic target of rapamycin complex 1 (mTORC1) response to stimuli including glucose, amino acids, and insulin at the single-cell resolution. The authors found that mTORC1 activation is graded, which supports the model of continuous adjustment of mTORC1 signaling to changes in the cellular environment rather than an "on/off" mode of mTORC1 function. Considering the pivotal role of mTORC1 in sensing a plethora of signals to orchestrate metabolic programs impacting cell fate and organismal physiology, it was thought that this study will be of interest across a broad spectrum of biomedical disciplines.

## Introduction

To adapt to continuously changing microenvironments, cells must perceive the availability of multiple nutrients and adjust anabolic and catabolic processes accordingly. The kinase mTOR (mammalian Target Of Rapamycin) is the core of a regulatory network for sensing growth factors and metabolites, including amino acids, glucose, and cellular energy charge (ATP/ADP/AMP ratios). Collectively, these factors regulate the assembly and activity of mTOR Complex 1 (mTORC1), a heteromer of mTOR, Raptor, mLST8, PRAS40, and DEPTOR (*Condon and Sabatini, 2019*). Through both transcriptional control and direct protein modification, the kinase activity of this complex coordinates many downstream processes, including protein, nucleotide, and lipid synthesis (*Shimobayashi and Hall, 2014*).

mTORC1 also regulates autophagy, the process of nutrient salvaging, and organelle biogenesis, via the kinase ULK1 and the transcription factor TFEB (*Kim et al., 2011*; *Settembre et al., 2011*).

While mTORC1 regulates many essential processes in the cell, its activity is not required to sustain these processes at basal rates (*Valvezan and Manning, 2019*). Cell lines lacking mTORC1 function can proliferate under *in vitro* conditions, where nutrients are in excess (*Cybulski et al., 2012*). However, mice die early in embryogenesis if they are either deficient for mTORC1 (*Guertin et al., 2006*) or maintain inappropriately high mTORC1 activity (*Efeyan et al., 2013*), indicating that mTORC1 regulation is needed under physiological conditions. These observations suggest that the essential physiological role of mTORC1 is to act as a controller that aligns the rates of catabolic and anabolic processes within a cell, with similarities to an engineered homeostatic control loop (*Valvezan and Manning, 2019*). However, current measurements of this controller's function are insufficient to establish a viable quantitative model. Extensive work in homeostatic control across many fields has shown that understanding a controller's function requires extensive characterization of its quantitative responses across its full range of operating conditions (*Doyle et al., 1992*). While many molecular details of mTORC1 activation have been elucidated, its response characteristics are sparsely mapped.

Regulation of mTORC1 activity is achieved spatially by the recruitment of mTORC1 and various adaptor components to the outer surface of the lysosomal membrane (*Menon et al., 2014*). When amino acids are abundant, the Ragulator-Rag complex dynamically scaffolds mTORC1 at the lysosome (*Lawrence et al., 2018*), where Rheb, a lysosome-associated small GTPase, activates mTORC1 catalytic activity. Growth factors exert their effect on mTORC1 through regulation of Rheb by modulating the localization of the TSC1/2 complex, a GTPase-activating protein (GAP), at the lysosomal membrane. Phosphorylation of TSC1/2 by the growth factor-induced kinases ERK and AKT promotes its dissociation from the lysosome, permitting Rheb to better activate mTORC1 (*Menon et al., 2014*). The energy-sensitive kinase AMPK can inhibit mTORC1 activity by both phosphorylating TSC1/2 to prevent its dissociation (*Inoki et al., 2003*) and by directly phosphorylating Raptor to inhibit mTORC1 assembly (*Gwinn et al., 2008*). Thus, at the molecular level mTORC1 activation resembles the function of a digital 'AND' gate, where an individual mTORC1 complex is only 'active' when mTORC1 can assemble *and* scaffold to the Ragulator-Rag complex *and* a nearby Rheb is active. This is much more likely when both amino acids and growth factors have been sensed and cellular energy status is sufficiently favorable. However, the cellular average of mTORC1 activity is expected to vary continuously (i.e. more akin to a rheostat) as mTOR and its regulators are present in $>10^3$ copies per cell and can cycle rapidly between multiple states within any individual cell (*Lawrence et al., 2018*).

Two major factors limit the ability of previous studies to evaluate the operation of mTORC1 within cells. First, past studies have not been able to resolve changes in mTORC1 activity over time. Most studies have used population averaging methods (such as immunoblots) or fixed single-cell snapshots (such as immunofluorescence), which can obscure the heterogeneous activity of single cells (*Birtwistle et al., 2012*; *Purvis and Lahav, 2013*). Averaging helps to resolve main effects by mitigating cell-to-cell heterogeneity, but prevents observation of true progression within individual cells (the functional context for mTORC1). In an extreme example, the average response to two different stimuli could arise from one subset of cells being activated by one stimulus, while different cells are activated by the other, and no cell actually responds to both stimuli. Snapshots can resolve this uncertainty as they capture heterogeneity and true states of single cells, but they cannot reliably link pre- and post-stimulus cells or determine how quickly a signal changes over time. Compounding this situation, when multiple stimuli are included in studies of mTORC1, they are typically applied simultaneously, but this represents only a subset of the contexts in which a cell may receive stimuli. Addressing the spectrum of feasible system behaviors requires non-destructive time-series measurements at single-cell resolution, with varied stimulus conditions. Second, mTORC1 activity is usually quantified by measuring phosphorylation of select substrates, including S6K (or its substrate, ribosomal protein S6), 4E-BP1, or ULK1 using phospho-specific antibodies. The measurements may be of high quality, but they are not guaranteed to be either linearly proportional to mTORC1 activity or even comparable across targets and studies as each substrate may differ in sensitivity both to mTORC1 and to additional regulators (*Kang et al., 2013*).

A potential strategy to more reliably quantify mTORC1 activity in live cells involves the transcription factor TFEB, a member of the microphthalmia (MiTF-TFE) family of transcription factors (*Goding and Arnheiter, 2019*). TFEB modulates the expression of genes required for lysosomal and mitochondrial

biogenesis (*Mansueto et al., 2017*; *Settembre et al., 2013*). Like many transcription factors, TFEB is regulated at the level of subcellular localization. mTORC1 phosphorylates TFEB at serines 122 and 211; phosphorylation at Ser-211 promotes binding to 14-3-3 proteins and sequestration of TFEB in the cytosol where it cannot access DNA (*Roczniak-Ferguson et al., 2012*; *Settembre et al., 2012*). When not phosphorylated at these serines, TFEB accumulates in the nucleus, where it activates the transcription of its target genes (*Sardiello et al., 2009*). The cytosolic-to-nuclear ratio of TFEB (hereafter $TFEB_{C/N}$) has been frequently used as a surrogate measure of this phosphorylation, both by immunofluorescence (*Marin Zapata et al., 2016*) and by live-cell microscopy using a fluorescent protein (FP) fusion tag (*Li et al., 2018*). However, as with many substrates, $TFEB_{C/N}$ is also regulated by phosphorylation at additional sites. GSK3β is a potent regulator of $TFEB_{C/N}$, phosphorylating Ser-134 and Ser-138 to enhance nuclear export (*Li et al., 2018*; *Li et al., 2016*). ERK and AKT have also been reported to phosphorylate TFEB (*Palmieri et al., 2017*; *Settembre et al., 2012*), with ambiguous impact on translocation. Using $TFEB_{C/N}$ as a live-cell readout for mTORC1 activity depends on disentangling these complicating factors.

In this study, we set out to characterize the homeostatic control function of mTORC1 at the cellular level. We used engineered versions of TFEB and immunostaining of multiple endogenous substrates to quantify the individual and combined effect of multiple inputs on mTORC1 activity at the single-cell level. Our data show that the response to amino acids and insulin varies among mTORC1 substrates, changes continuously and dose-dependently with both stimuli, and also continues adapting on the order of hours after major stimuli. With this knowledge in hand, we were able to use multiplexed immunofluorescence data to generate quantitative models of protein synthesis as a function of mTORC1 activity. These models revealed a degree of control at the single-cell level that is consistent with a homeostatic role for mTORC1, but which is strikingly distributed across the mTORC1 signaling network.

## Results

### Fluorescent reporters for TFEB localization enable continuous monitoring of mTORC1 activity in living cells

To investigate mTORC1 activity with high temporal resolution, we examined TFEB localization in live cells. The use of TFEB as a reporter for mTORC1 is complicated by non-mTORC1-specific aspects of its regulation, including sites potentially phosphorylated by GSK3β and AKT (*Li et al., 2016*; *Palmieri et al., 2017*). We therefore engineered multiple versions of TFEB fused to a fluorescent protein and expressed them stably in MCF10A cells (*Figure 1A*). We first compared full-length TFEB fused to mVenus (FL-TFEB-TR) to a truncated version (TFEB-TR) comprising residues 1–237, which lacks both the DNA-binding domain and a C-terminal AKT phosphorylation site. An advantage of this truncation is that it reduces the deleterious effects of overexpressing TFEB, which were evident in altered cellular morphology and low expression level of FL-TFEB-TR in stable clones. We also constructed phospho-site mutants to decouple GSK3β and mTORC1 regulation of TFEB. In 3xSA-TR, serines at positions 134, 138, and 142, which are involved in GSK3β and PKC control of $TFEB_{C/N}$ (*Li et al., 2018*; *Li et al., 2016*), were converted to alanine. In 5xSA-TR, the aforementioned sites and the mTORC1 sites Ser-122 and Ser-211 were all converted to alanine to yield a reporter that is expected to be unresponsive to both GSK3β and mTORC1. Finally, as a parallel measurement of mTORC1-dependent translation, we constructed cell lines carrying TOP-H2B-YFP-DD (*Han et al., 2014*). This reporter uses a chemically inhibited degron to quantify the translation rate of an mRNA carrying a 5′ terminal oligo-pyrimidine (TOP) motif, which is found in many mRNAs regulated by mTORC1 activity (*Thoreen et al., 2012*).

We exposed cells to conditions with various concentrations of nutrients, including glucose, glutamine, insulin, and essential amino acids (measured 'x'-fold relative to standard media concentrations – for specific values, see manufacturer's specifications). For all reporters, we quantified responses to treatment on a cell-by-cell basis using a metric, Δi, which computes the average change in reporter localization from baseline to the peak response, typically 30–60 min after treatment. Using this quantification, we observed that FL-TFEB-TR$_{C/N}$ and TFEB-TR$_{C/N}$ showed similar responses to a panel of mTORC1 modulators (*Figure 1B, C and F* and *Figure 1—figure supplement 1A and B*), including the inhibitors Torin2 and rapamycin. Notably, in previous work, induction of TFEB nuclear localization by rapamycin was below the limit of detection based on either immunofluorescence or static images of

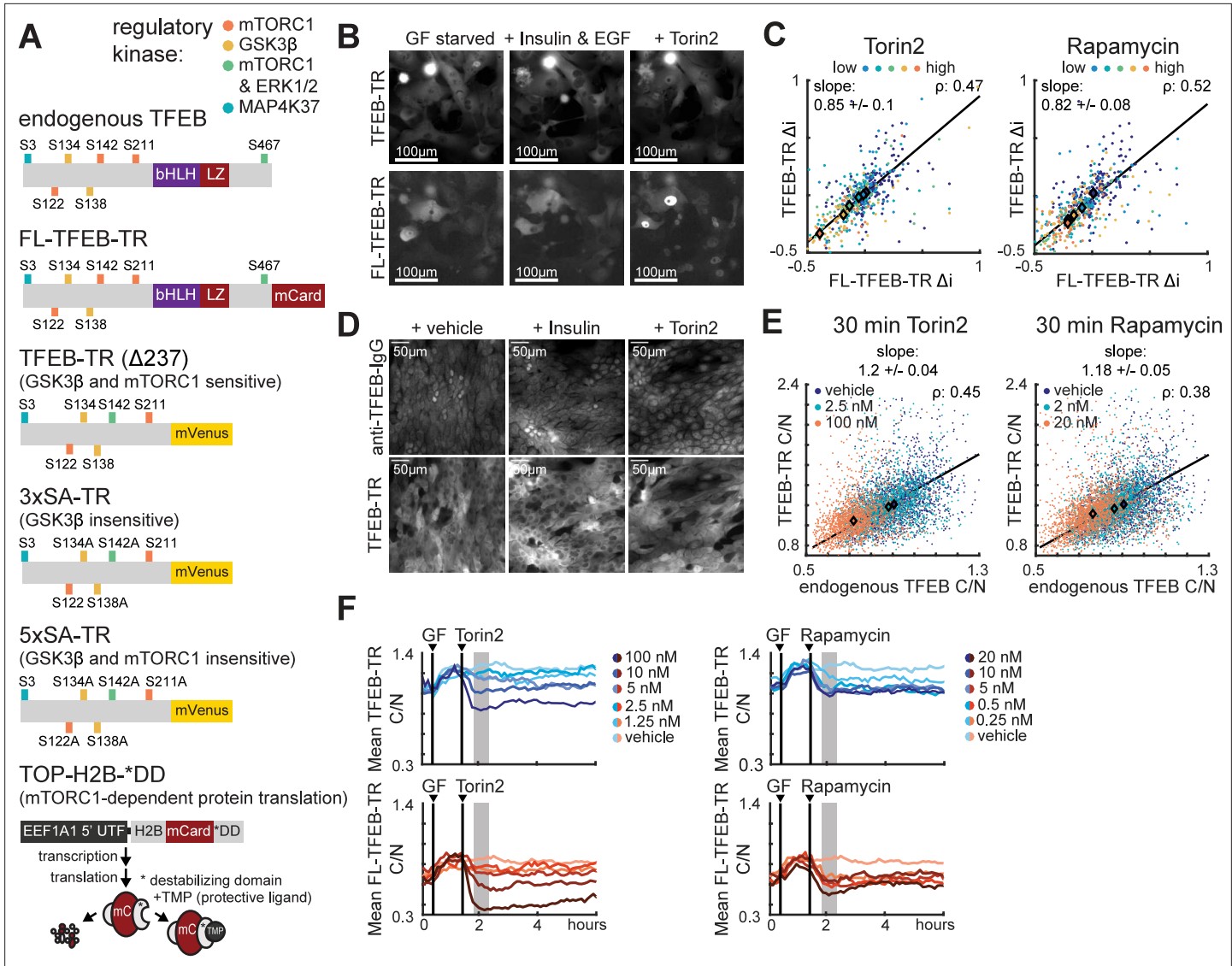

**Figure 1.** Fluorescent reporters for TFEB localization enable monitoring of mTORC1 activity in living cells. (**A**) Schematic of fluorescent protein reporters used in this study. From top to bottom, regulatory phosphorylation sites are indicated for endogenous TFEB, full-length TFEB translocation reporter (FL-TFEB-TR), truncated TFEB-TR (amino acids 1–237), 3xSA-TR (lacking potential GSK3β target sites), and 5xSA-TR (lacking both GSKβ- and mTORC1 sites). The TOP-H2B-DD reporter (bottom) measures mTORC1-dependent protein translation as the rate of fluorescence increase following inhibition of degradation with TMP (*Han et al., 2014*). (**B**) Microscopy images of MCF10A cells expressing both FL-TFEB-TR and TFEB-TR. Cells were first starved of growth factors for 6 hr, then treated for 1 hr with vehicle, EGF (20 ng/mL) and insulin (100 ng/mL), or Torin2 (100 nM). Representative images from two experiments are shown. Images represent an arbitrarily selected subset of the full plotted dataset. (**C**) Scatter plots of same-cell Δi values for dual reporter MCF10A cells (**B**) treated with Torin2 or rapamycin titrations, for >150 cells per condition. Δi is defined as the change over a 1 hr period following treatment. The Pearson correlation ($\rho$) for same-cell FL-TFEB-TR and FL-TFEB-TR Δi TFEB-TR is indicated, along with a linear fit of the relationship and its slope (*Figure 1—source data 1*). (**D**) Microscopy images showing immunofluorescence detection of endogenous TFEB in TFEB-TR-expressing MCF10A cells. Cells were starved as in (**B**) and treated with either insulin or Torin2 and fixed following 30 min of treatment. Endogenous TFEB was detected with an antibody against the C-terminal region that is absent in TFEB-TR. (**E**) Scatter plot of same-cell C/N ratios for endogenous TFEB and TFEB-TR (**D**), representing >600 cells per condition, presented as in (**C**) (*Figure 1—source data 2*). (**F**) Comparison of temporal TFEB-TR and FL-TFEB-TR localization responses. The mean C/N ratio for each reporter is shown for MCF10A cells expressing both reporters. Treatments with 20 ng/mL EGF and 10 µg/mL insulin (GF), followed by Torin2 or rapamycin, are indicated by vertical lines. Gray regions denote the area for Δi calculation. Averages were calculated from >150 cells, with experiments performed in duplicate.

The online version of this article includes the following source data and figure supplement(s) for figure 1:

**Source data 1.** TFEB-TR compared with full-length TFEB-TR, live-cell data.

**Source data 2.** TFEB-TR compared with endogenous TFEB, immunofluorescence data.

**Figure supplement 1.** TFEB reporter validations.

TFEB-GFP (*Roczniak-Ferguson et al., 2012*; *Settembre et al., 2012*). The ability to detect responses to even low doses of rapamycin (5 nM) indicates the enhanced sensitivity that is possible using current image quantification methods and time-resolved data, which allow direct comparison of the drug-treated state to the pretreated baseline for each cell. Similarly, the response to insulin, which was previously undetected when fixed time points were analyzed (*Napolitano et al., 2020*), was clearly detectable and significant using Δi as a metric (*Figure 1—figure supplement 1A*). FL-TFEB-TR localization was more biased toward the nucleus than TFEB-TR, but when expressed in the same cell, both reporters showed similar dynamic changes in C/N ratio and showed high similarity by cross-correlation analysis (*Figure 1F* and *Figure 1—figure supplement 1D and E*). Based on this similarity and the reduced impact on cellular behavior, we subsequently focused on TFEB-TR.

Localization of TFEB-TR was compared to endogenous TFEB on a cell-by-cell basis using immunostaining with an antibody targeting the C-terminal region of TFEB that is absent in TFEB-TR (*Figure 1D*). As with FL-TFEB-TR, endogenous TFEB localized more strongly in the nucleus than TFEB-TR did, but a linear relationship was retained between the C/N ratios of TFEB-TR and endogenous TFEB over a range of kinase inhibitor treatments (*Figure 1E* and *Figure 1—figure supplement 1C*). Thus, TFEB-TR represents changes in the subcellular distribution of endogenous TFEB, without evidence of saturation. Although TFEB phosphorylation has been reported to alter TFEB protein stability (*Sha et al., 2017*), total TFEB-TR fluorescence changed on a slower time scale than changes in localization, suggesting that localization is the parameter most sensitive to time-dependent changes in mTORC1

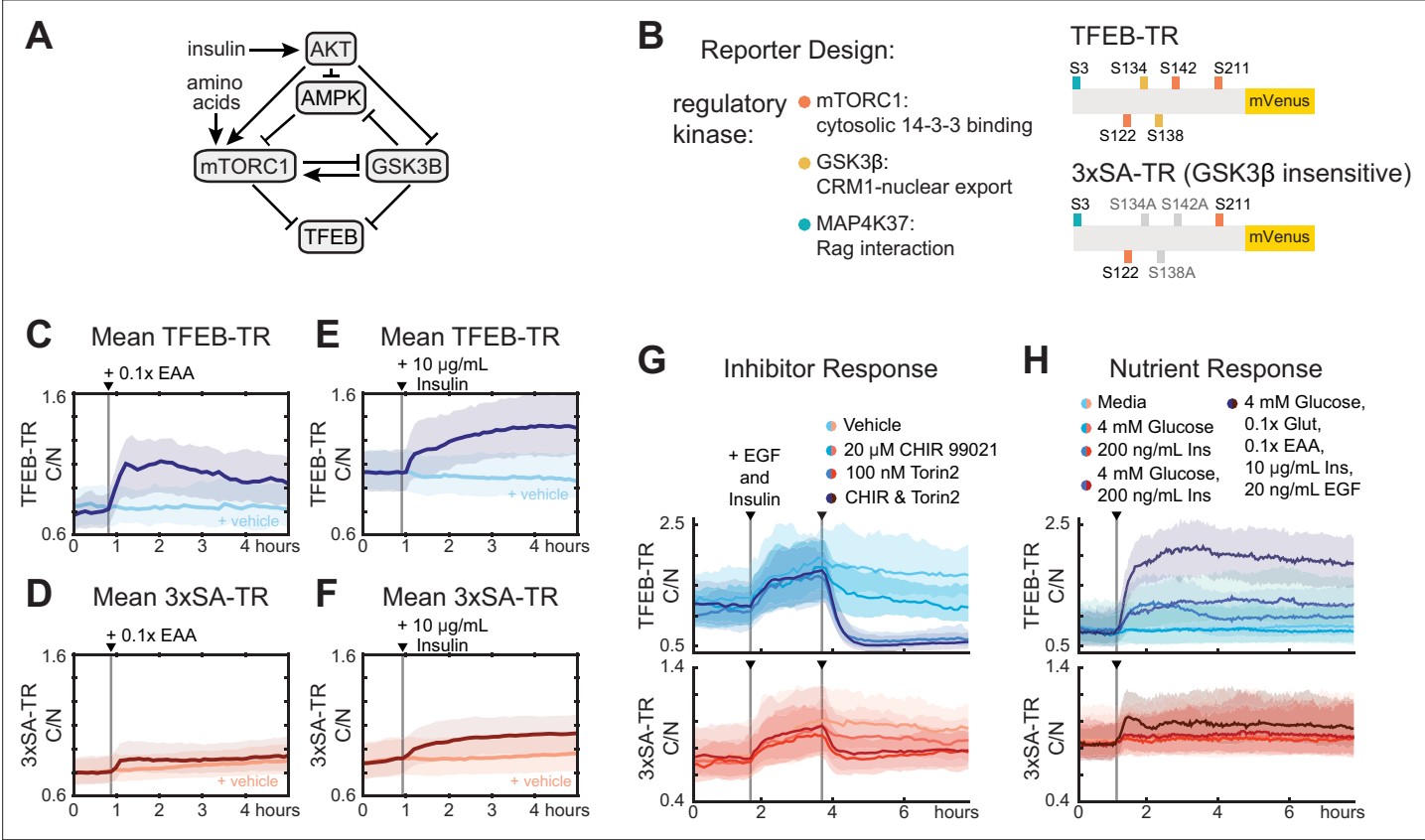

**Figure 2.** Phosphorylation by GSKβ amplifies cytosolic to nuclear ratio of TFEB. (**A**) Schematic diagram of kinase interactions regulating TFEB nuclear localization. (**B**) Illustration of wild-type TFEB-TR, and the GSKβ-insensitive reporter, 3xSA-TR. (**C–F**) Temporal changes in reporter C/N ratio following stimulation with essential amino acids (EAA, 0.1×) or insulin (10 µg/mL), shown alongside a vehicle control. Experiments were performed in MCF10A cells expressing both TFEB-TR and 3xSA-TR. (**G, H**) Comparison of nutrient and inhibitor responses for both TFEB-TR and 3xSA-TR reporters, in dual reporter cells. Mean C/N ratio is indicated by bold lines with the 25th and 75th percentiles indicated by the shaded regions (*Figure 2—source data 1*).

The online version of this article includes the following source data and figure supplement(s) for figure 2:

**Source data 1.** TFEB-TR compared with 3xSA-TR, live-cell data, nutrient, and inhibitor treatments.

**Figure supplement 1.** Characterization of TFEB-TR desensitizing mutations.

activity (*Figure 1—figure supplement 1F and G*). We also tested whether reporter expression alters the kinetics of mTORC1 substrate phosphorylation by measuring pS6 or p4E-BP1 in a time course, and we observed no difference in average phosphorylation kinetics between reporter-expressing and parental MCF10A cells (*Figure 1—figure supplement 1H*). We conclude that TFEB-TR$_{C/N}$ tracks with endogenous TFEB$_{C/N}$ in live cells in response to both amino acid and insulin stimuli, with minimal perturbation of endogenous mTORC1 signaling.

## Phosphorylation of TFEB by GSK3 amplifies cytosolic localization induced by mTORC1

We next investigated the role of GSK3β in the regulation of TFEB, which involves both direct and indirect effects on TFEB localization and may exert opposing effects on TFEB downstream of AKT (*Li et al., 2018*; *Li et al., 2016*; *Figure 2A*). To assess the importance of GSK3β phosphorylation of TFEB, we compared TFEB-TR to 3xSA-TR (*Figure 2B*) across a range of conditions (*Figure 2C–H*). Consistent with the reported role of the mutated serines in Crm1-mediated nuclear export of TFEB (*Li et al., 2018*), the average C/N ratio was decreased for 3xSA-TR across all conditions. Like TFEB-TR, 3xSA-TR responded to amino acids and insulin, but did so with a reduced amplitude (*Figure 2D and F* and *Figure 2—figure supplement 1D and E*). C/N ratio decreased for both reporters in response to direct inhibitors of mTOR. This decrease was large relative to baseline for TFEB-TR$_{C/N}$, whereas mTOR inhibition only returned 3xSA-TR$_{C/N}$ to its baseline, indicating a reduction in its overall dynamic range (*Figure 2G and H*). To quantify this decrease in dynamic range, we directly compared Δi for the two reporters expressed in the same cell (*Figure 2—figure supplement 1C–F*). Correlation between the reporters was strong given the expected variance in single-cell measurements. However, the slopes of the relationships were considerably lower (0.21, 0.26) than the slope of 1 expected for equivalent responses. These slopes indicate a smaller change in 3xSA-TR$_{C/N}$ than in TFEB-TR$_{C/N}$ across all conditions. Taken together, these findings indicate that phosphorylation of TFEB by GSK3β expands the dynamic range of TFEB translocation in response to both increases and decreases in mTORC1 activity.

Surprisingly, despite the mutation of its GSK3β phosphorylation sites, 3xSA-TR showed a decrease in C/N ratio in response to a GSK3β inhibitor (CHIR99021, *Figure 2G*). However, a possible explanation is indirect inhibition of mTORC1 activity through known regulation by GSK3β (*Stretton et al., 2015*), and consistent with this, CHIR99021 decreased staining for pS6 under a range of conditions (*Figure 2—figure supplement 1A*). As expected, all responsiveness to insulin and amino acids was lost in the 5xSA-TR construct, in which the two remaining sites required for mTORC1-dependent responses (Ser-122 and Ser-211) were mutated (*Figure 2—figure supplement 1B*).

## Graded sensing of multiple stimuli by mTORC1-TFEB

Having characterized the TFEB-TR and 3XSA-TR reporters, we used them to characterize how the mTORC1-TFEB pathway senses increasing concentrations of individual stimuli. This approach distinguishes sharp 'digital' responses, which appear as bimodal distributions at intermediate stimulus concentrations, from 'analog' or graded responses, which are characterized by a gradual shift in the mean of a unimodal distribution (*Tyson et al., 2003*; *Figure 3A*). Distributions of individual cell responses were measured by first withdrawing amino acids, and then calculating the Δi (peak change in the first hour) or Δss (change after 3–4 hr) for each individual cell (*Figure 3B*) after amino acids were resupplied. Both metrics shifted gradually and unimodally with the concentration of amino acids, indicating that the individual cell response to amino acids is graded. Similar distributions were also obtained for 3xSA-TR and TOP-H2B-YFP-DD (*Figure 3C*), and graded responses were also observed for other mTORC1 modulators, including insulin, and pharmacological inhibitors of mTORC1 and its upstream activators (*Figure 3D and E* and *Figure 3—figure supplement 1*). Notably, the average TFEB-TR localization in cells cultured without insulin is consistent with that when treated with the AKT inhibitor MK2206, indicating that baseline AKT activity is low in these cells in the absence of insulin (*Figure 3B and D* and *Figure 3—figure supplement 1C*). Together, these results demonstrate that, in general, the response of mTORC1-TFEB signaling is continuously graded.

To evaluate how multiple inputs interact in the mTORC1-TFEB pathway, we took advantage of the fact that with a live-cell reporter stimulus responses can be measured sequentially in the same cell. We developed a protocol in which cells were cultured in the absence of sugars, amino acids, and growth factors, which were then replaced in various sequences at 2 hr intervals (*Figure 4*). As internal controls

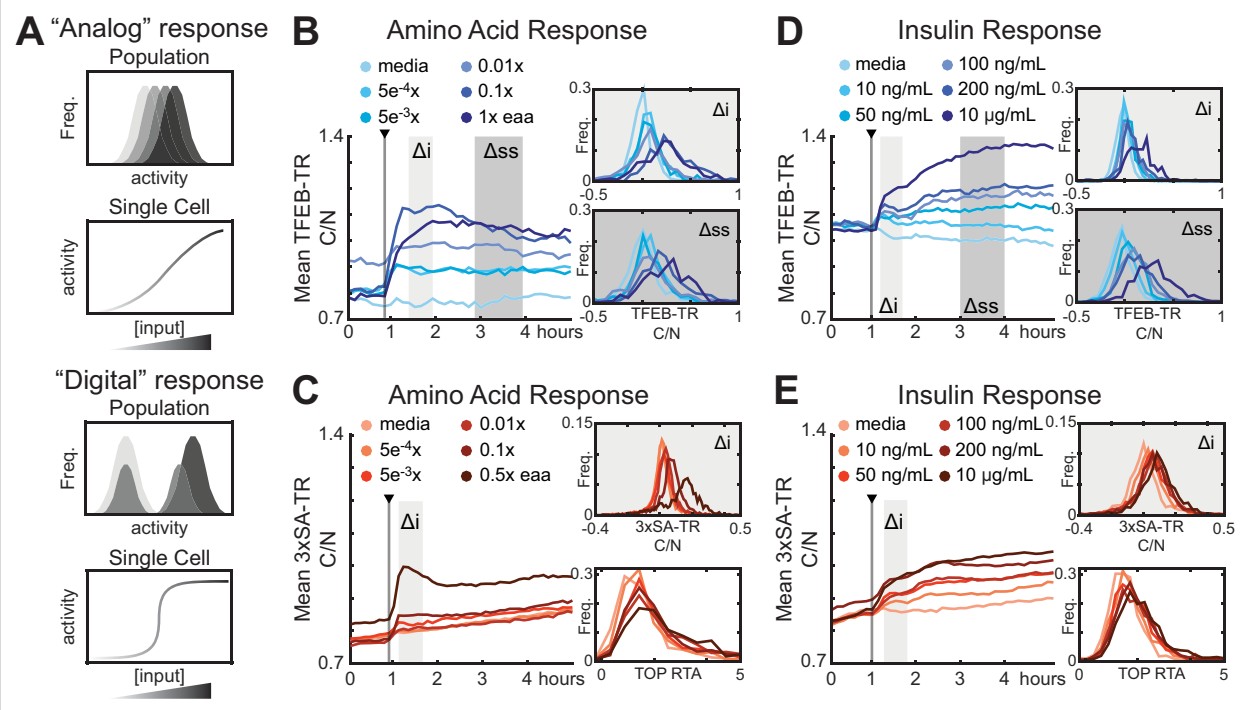

**Figure 3.** Analog regulation of mTORC1-TFEB axis by input strength. (**A**) Conceptual comparison of 'Analog' and 'Digital' signal responses to graded input concentrations. Histograms and single-cell plots represent the expected results if individual cells show a dose–responsive (analog) behavior or a sharply switching (digital) behavior at a threshold concentration. (**B–E**) Response of MCF10A cells expressing TFEB-TR, 3xSA-TR, or H2B-TOP-DD to titrations of mTORC1 modulators. The mean C/N ratio over time for over >500 cells is reported as a line, alongside histograms of the initial (Δi) and steady-state (Δss) responses for individual cells. Histograms showing the relative translation activity (RTA), as measured by the TOP reporter, from a parallel experiment are show in place of Δss in (**C, E**). (**B, C**) Cells were starved of growth factors and amino acids for 6 hr, then stimulated with essential amino acids. (**D, E**) Cells were starved of growth factors for 6 hr, then treated with titrations of insulin (*Figure 3—source data 1*).

The online version of this article includes the following source data and figure supplement(s) for figure 3:

**Source data 1.** TFEB-TR and 3xSA-TR, live-cell data, insulin and amino acid treatments.

**Figure supplement 1.** Quantification of responses of TFEB-TR, 3xSA-TR, or H2B-TOP-DD to titrations of mTORC1 modulators, as in *Figure 3B–E*.

for the activity of upstream mTORC1 regulators, we included reporters for AKT and AMPK (*Hung et al., 2017*). When glucose and glutamine were added first (*Figure 4A*), an immediate increase in mean TFEB-TR$_{C/N}$ was observed, coinciding with a decrease in mean AMPK activity. Subsequent treatment with insulin resulted in a further increase in mean TFEB-TR$_{C/N}$, along with a corresponding activation of AKT. These responses align with basic expectations for mTORC1 regulation by TSC1/2 and Rheb. A third treatment, with amino acids, spurred another increase in mean TFEB-TR$_{C/N}$, with no further change in AMPK activity, and a small decrease in AKT activity, which is consistent with amino acids activating mTORC1 through a Rag-dependent mechanism independent of both AKT and AMPK.

Individual cells also displayed graded changes in TFEB-TR$_{C/N}$ in response to the sequential stimuli (*Figure 4F*). However, responses to different stimuli varied in magnitude, and in many cases were difficult to distinguish from the basal variability of TFEB-TR$_{C/N}$. These data confirm that individual cells can respond incrementally to multiple mTORC1 stimuli, but also demonstrate variation in sensitivity to each, which may represent underlying heterogeneity in the metabolism of individual cells.

When the sequence of stimuli was modified, we noted changes in the average size of TFEB-TR responses. Notably, when insulin was added before amino acids, glucose, or glutamine, there was a minimal response of TFEB-TR$_{C/N}$, despite a robust activation of AKT (*Figure 4B*). When glucose and glutamine were subsequently introduced, the resulting TFEB-TR$_{C/N}$ response reached a magnitude similar to that achieved by insulin following glucose and glutamine. This dependency may indicate that direct activation of mTORC1 by AKT-mediated phosphorylation is substantially weaker than by the AKT-stimulated uptake of glucose, which could directly activate mTORC1 and/or suppress AMPK activity. The activity profile for AMPK is consistent with this latter possibility. In contrast to this

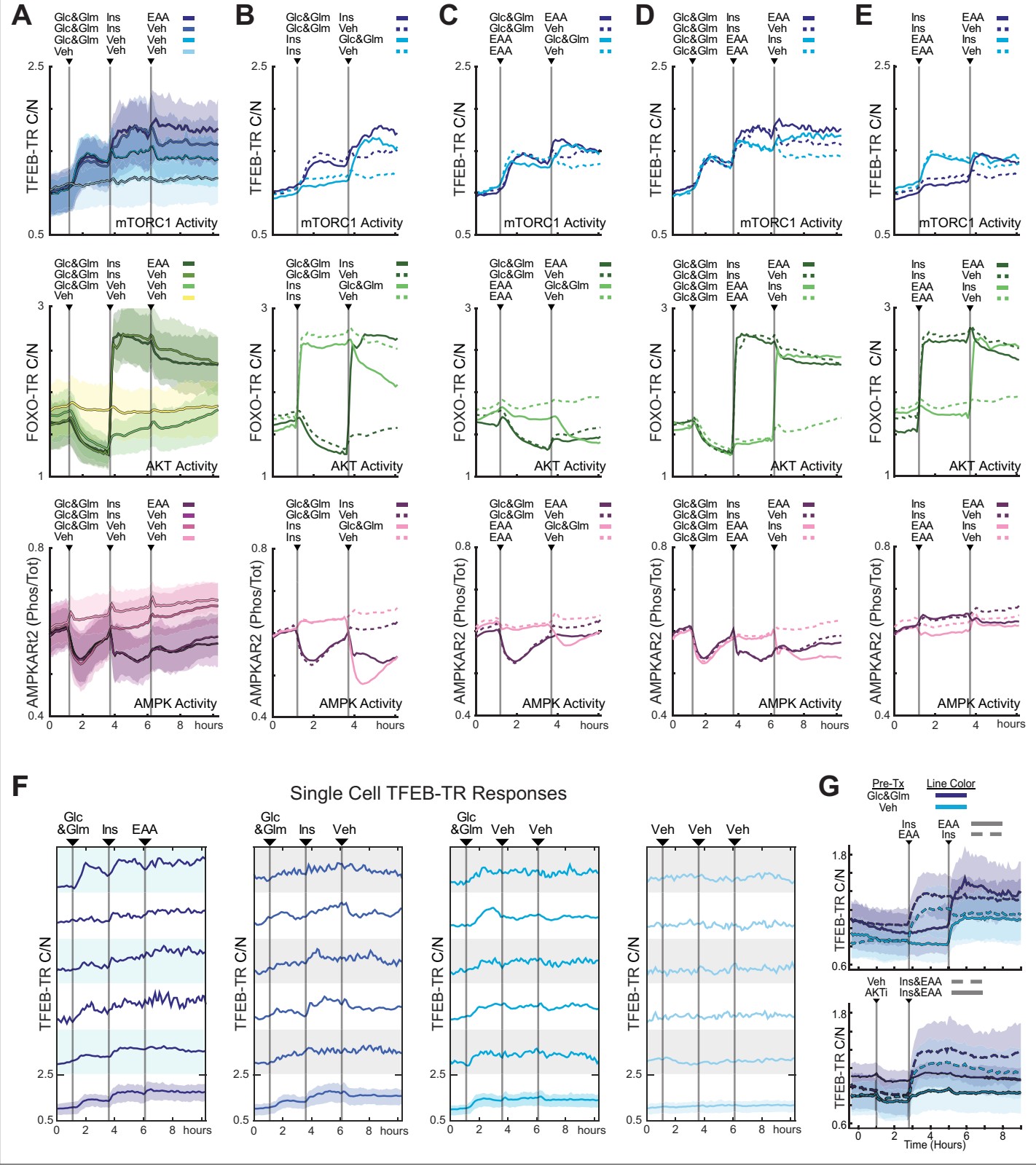

**Figure 4.** TFEB responds to sequential nutrient addition through incremental changes in localization. (**A**) Mean reporter responses to sequential nutrient stimulation in MCF10A cells expressing TFEB-TR, an AKT reporter (FOXO-TR), and an AMPK activity FRET reporter (AMPKAR2), calculated from at least 200 cells per treatment. Cells were starved of nutrients and growth factors for 6 hr, then stimulated with a sequence of glucose (17.5 mM) and glutamine (2.5 mM), followed by insulin (10 μg/mL), and then essential amino acids (0.1×), or vehicle controls as indicated. Shaded regions denote the

*Figure 4 continued on next page*

*Figure 4 continued*

25th/75th percentile range. (**B–E**) Mean reporter responses to various nutrient stimulation sequences in the MCF10A triple reporter cell line. Cells were treated as in (**A**), with the sequences indicated. Lines represent means of >200 cells; dotted lines indicate vehicle controls. (**F**) Single-cell responses of TFEB-TR C/N for the stimulus sequence shown in (**A**). Representative cells were chosen randomly, excluding severe outliers (*Figure 4—source data 1–5*) (**G**) Mean response of TFEB-TR reporter in mouse embryonic fibroblast (MEF) cells (displayed as in **A**), but with line color indicating pretreatment condition and dashed vs. solid lines indicating treatment sequence (*Figure 4—source data 6*).

The online version of this article includes the following source data and figure supplement(s) for figure 4:

**Source data 1.** TFEB-TR, FOXO-TR, AMPKAR2, live-cell data corresponding to *Figure 4A*.

**Source data 2.** TFEB-TR, FOXO-TR, AMPKAR2, live-cell data corresponding to *Figure 4B*.

**Source data 3.** TFEB-TR, FOXO-TR, AMPKAR2, live-cell data corresponding to *Figure 4C*.

**Source data 4.** TFEB-TR, FOXO-TR, AMPKAR2, live-cell data corresponding to *Figure 4D*.

**Source data 5.** TFEB-TR, FOXO-TR, AMPKAR2, live-cell data corresponding to *Figure 4E*.

**Source data 6.** TFEB-TR in wild-type mouse embryonic fibroblast (MEF) cells, live-cell data, insulin and amino acid treatments.

**Figure supplement 1.** Mean responses of 3xSA-TR to sequential nutrient stimulation in MCF10A cells, presented as in *Figure 4*.

dependency, a substantial TFEB-TR response to amino acids was observed even prior to glucose/glutamine addition (*Figure 4C*), indicating that Rag-mediated activation of mTORC1 is independent of extracellular glucose and glutamine. Finally, we alternated the order of amino acid and insulin stimuli in the presence (*Figure 4D*) and absence (*Figure 4E*) of glucose and glutamine. In both cases, TFEB-TR$_{C/N}$ responses to insulin and amino acids were consistent regardless of their order in the sequence; we did not observe any enhancement of insulin or amino acid response by pretreatment with the other factor. Rather, we observed some attenuation of the response for the last stimulus when all others were present, consistent with reaching the saturation region of either the reporter or mTORC1 itself. When the same experiments were repeated with the 3xSA-TR biosensor, results were consistent with those of TFEB-TR, but the diminished range of the 3xSA reporter reduced the ability to distinguish shifts in reporter localization from noise (*Figure 4—figure supplement 1*).

To explore TFEB reporter responses in a different cell type, we performed a similar experiment with TFEB-TR in wild-type mouse embryonic fibroblasts (MEFs). MEFs displayed a strong increase in TFEB-TR$_{C/N}$ when treated with amino acid stimuli, but unlike MCF10A cells had no detectable individual response to insulin, regardless of the presence of glucose and glutamine (*Figure 4G*). Similarly, pretreatment with an amino acid stimulus did not affect the TFEB-TR response to insulin. However, despite their unresponsiveness to insulin, AKT activity appeared to play a significant role in TFEB-TR localization, as pre-incubation with the AKT inhibitor MK-2206 strongly limited the amino acid response. Moreover, treatment with MK-2206 induced a small, but measurable and immediate, decrease in TFEB-TR$_{C/N}$, which suggests that baseline AKT activity is present in the absence of growth factor stimulation and contributes to TFEB-TR localization. These results are consistent with the interpretation that TFEB-TR generally provides a readout of mTORC1 activity as it adjusts to sequential amino acid and insulin stimuli.

## Temporal fluctuations in mTORC1 activity track individual cell metabolic states

When monitoring cells over time, we noted that TFEB-TR$_{C/N}$ did not remain constant following each stimulus. In many cases, it showed a rebound effect following an initial peak or a gradual rise across hours (e.g., see *Figure 4A and F*). These transients imply that mTORC1-TFEB signaling and cellular metabolism continually adjust to metabolite availability. We therefore tracked cells over a longer period of time (12 hr) under conditions of starvation and refeeding with essential amino acids at different concentrations (*Figure 5A*). Initially after refeeding, average TFEB-TR$_{C/N}$ changed sharply depending on the amino acid concentration. Subsequently, however, mean trajectories converged over the 12 hr after re-feeding, regardless of the amino acid concentration. Adaptation on this time scale indicates that mTORC1-TFEB signaling eventually reaches a new steady state, consistent with previous data (*Marin Zapata et al., 2016*) and the expectation that TFEB-directed induction of autophagy will eventually provide more amino acids, leading to resumed activity of mTORC1.

To observe this adaptation process at the single-cell level, we analyzed TFEB-TR$_{C/N}$ in individual cells under the same conditions. In media lacking amino acids, TFEB-TR$_{C/N}$ fluctuated asynchronously,

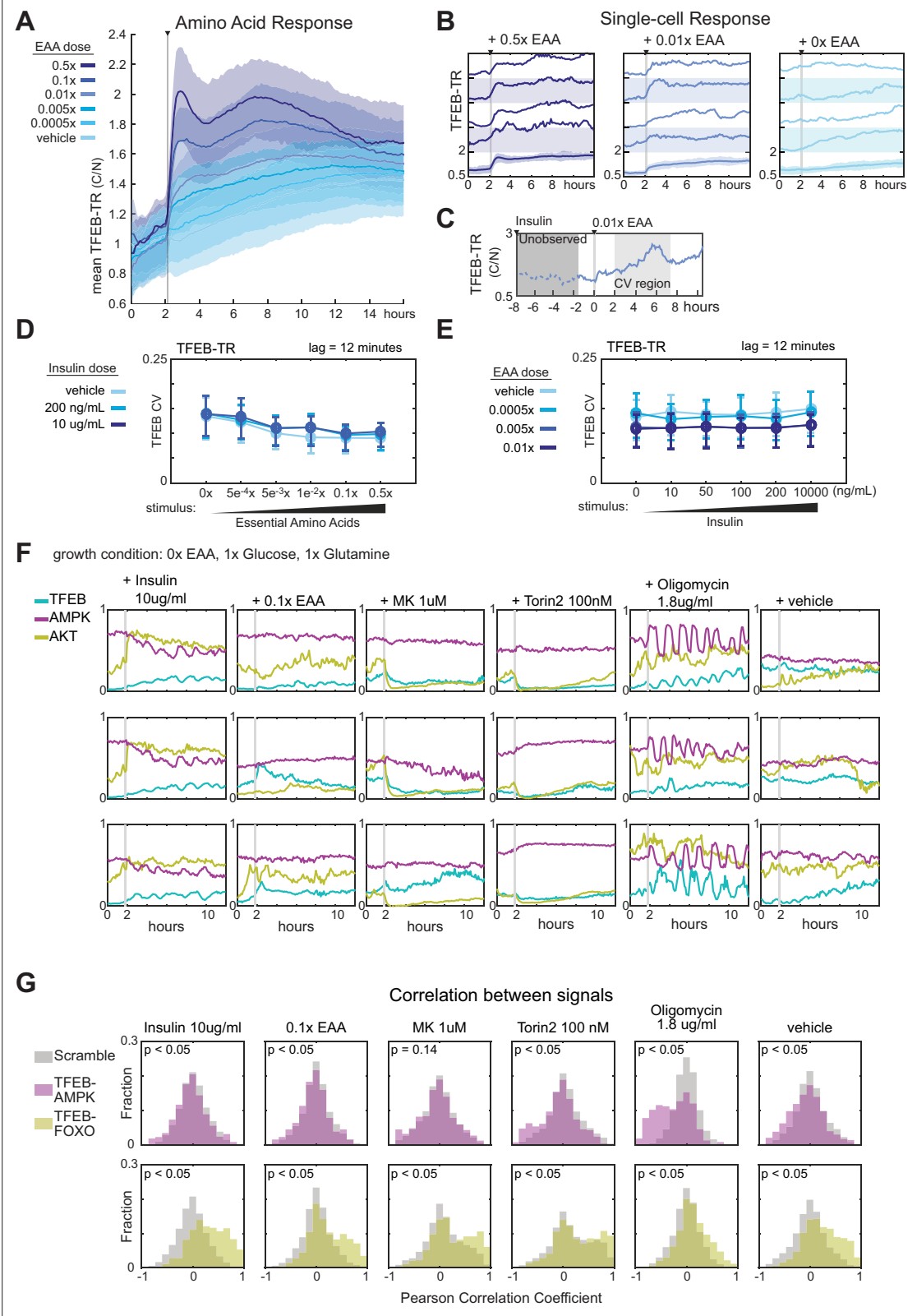

**Figure 5.** Nutrient limitation induces correlated fluctuations in the AMPK-mTORC1-TFEB network. (**A**) Mean TFEB-TR C/N response over 12 hr in MCF10A cells, following starvation of growth factors and amino acids for 6 hr and then stimulated with essential amino acids at a range of concentrations. Data from >400 cells in each condition. (**B**) Single-cell TFEB-TR C/N responses, for several conditions corresponding to the experiment in (**A**). The bottom traces show the mean with shaded regions denoting the 25th and 75th percentiles. (**C**) Schematic illustrating an example experiment

*Figure 5 continued on next page*

*Figure 5 continued*

timeline and time window used for coefficient of variation calculations in (**D**) and (**E**). (**D, E**) Single-cell coefficient of variation (CV) in TFEB-TR C/N for MCF10A cells grown in dual treatments of insulin and essential amino acids. CV was calculated for >400 individual cells per condition, across 2–7 hr after stimulation, with point-to-point noise reduced by subsampling to 12 min intervals. Error bars report the interquartile range of the CV. (**D**) Cells were cultured in insulin for 6 hr then stimulated with amino acids. (**E**) Cells were cultured in amino acids for 6 hr then stimulated with insulin (*Figure 1—source data 1 and 2*) (**F**) Single-cell traces for triple reporter MCF10A cells. Cells were starved of amino acids for 6 hr and then treated with insulin, essential amino acids, MK2206, Torin2, Oligomycin, or a vehicle control. (**G**) Correlation of single-cell TFEB-TR and AMPKAR activity (upper) or TFEB-TR and FOXO-TR activity (lower), calculated for >150 cells during an 8 hr period following the indicated treatments. Optimal lags were identified by cross-correlation, and the median lag used to plot correlation distributions. All median lags were zero. Gray bars indicate the distribution of 'scrambled' controls in which reporter signals from different cells were paired randomly (*Figure 5—source data 4*).

The online version of this article includes the following source data and figure supplement(s) for figure 5:

**Source data 1.** TFEB-TR response to insulin and amino acid combinations without glucose, live-cell data.

**Source data 2.** TFEB-TR response to amino acids after insulin treatment, with glucose, live-cell data.

**Source data 3.** TFEB-TR response to insulin after amino acid treatment, with glucose, live-cell data.

**Source data 4.** TFEB-TR, AMPKAR, FOXO-TR response to stimuli and inhibitors, live-cell data.

**Figure supplement 1.** Extended view of AMPK-mTORC1-TFEB fluctuations.

with a time scale of approximately 2 hr (*Figure 5B* and *Figure 5—figure supplement 1A*). Unlike pulsatile signals observed in some other pathways (*Albeck et al., 2013*; *Lahav et al., 2004*), TFEB-TR$_{C/N}$ fluctuations were not sharply defined. Therefore, to quantify fluctuations, we calculated the coefficient of variation for each individual trajectory, over hours 1–6 after the amino acid spike. These calculations identified a trend in which the coefficient of variance (CV) per cell was reduced under increasing extracellular amino acid concentrations, such that cells showed the largest fluctuations in TFEB-TR$_{C/N}$ when amino acids are limited (*Figure 5C–E*). In contrast, fluctuations remained similar across all concentrations of insulin. We infer that the abundant availability of amino acids precludes the need for adjustments to limiting mTORC1 activity, but under more physiological conditions (<0.01×) nutrient stress is more common, rendering mTORC1 more sensitive to its regulators.

To understand the sources of time-dependent variation in the mTORC1 signaling network, we monitored TFEB-TR$_{C/N}$ correlation with AKT and AMPK activities in individual cells over time. We noted no trend in CV for FOXO-TR, and an increase in AMPKAR2 CV with increasing insulin (*Figure 5—figure supplement 1B–E*). Over an array of conditions designed to modulate mTORC1 input pathways, we observed correlated fluctuations in these activities (*Figure 5F*). Oligomycin, which induces energetic stress and large pulses in AMPK activity (*Kosaisawe et al., 2021*), provoked the most pronounced and apparently correlated activity changes in TFEB-TR, AMPK, and AKT. However, other conditions also produced fluctuations with visible relationships between the kinase activities. Cross-correlation analysis was performed to quantify these relationships between AKT and TFEB-TR, and between AMPK and TFEB-TR, revealing significant cellular co-variation (*Figure 5G*). Together, this analysis demonstrates that AKT, AMPK, and mTORC1 activity fluctuations are coupled at the cellular level, with the largest fluctuations (other than those produced by inhibitors) observed when amino acids are limited. These data reinforce that mTORC1 activity is continually and rapidly responsive to changes within its network of input pathways.

## mTORC1 network phosphorylation states in single cells

Having clarified the time-dependent dynamics of mTORC1 via TFEB-TR, we questioned whether other mTORC1 substrates respond differently during dynamic activity changes, generating a dataset of quantitative cyclic immunofluorescence measurements for a panel of endogenous mTORC1 targets, including 4E-BP1 (phosphorylation at Thr-37/46), ULK1 (phosphorylation at Ser-757), S6K (phosphorylation at Thr-389), S6 (phosphorylation at Ser-240/244 or at Ser-235/236 by the direct mTORC1 target S6K1/2; note that Ser-235/236 is also regulated independently of mTOR by p90-Rsk), and TFEB (cytosolic/nuclear ratio), as well as related proteins including phosphorylated AKT, GSK3β, and ACC. Cells were cultured in minimal medium and fixed 1 hr after the introduction of stimulation conditions. We assessed protein synthesis activity in each case by pulsing cells with O-propargyl puromycin (OPP), which is incorporated into translating peptides, for 30 min prior to fixation, followed by fluorescence labeling with click chemistry. All stains were

performed on the same samples using a bleach/stain cycle to yield a total of 16 measurements for each cell. We performed target staining under 16 stimulation conditions, comprising all present/ absent combinations of saturating insulin, essential amino acids (at a concentration inducing the maximal effect), glucose, and glutamine. Over $10^4$ cells per condition were quantified to obtain well-sampled distributions (*Figure 6A* and *Figure 6—figure supplement 1*). The measured cellular staining intensities followed expected patterns of activity, with the combined stimuli eliciting the greatest phosphorylation of mTORC1 targets. Across the experiment, stain intensities yielded unimodal near-normal distributions, though pS6 (both 235/236 and 240/244) exhibited markedly skewed distributions under low to moderate stimulation conditions (*Figure 6A*), which were visible as highly variable individual cell intensities (*Figure 6—figure supplement 1*). The stains generally followed expected patterns based on treatment conditions; for example, pAKT and pGSK3-beta were found primarily in the insulin-treated cells, while pACC was found predominantly in the most nutrient-deprived cells.

As a qualitative visualization of network activity at the single-cell level, we constructed UMAPs based on the known mTORC1 targets and their substrates (pS6K1, p4E-BP1, pS6-235/236, pS6-240/244, pULK1, and TFEB$_{C/N}$; *Figure 6B*). This projection visualizes the approximate relationships of mTORC1 target phosphorylation and the interrelation of different treatment conditions within this space (*Figure 6—figure supplement 2*). Consistent with the concept that mTORC1 targets vary in their affinity for phosphorylation (*Kang et al., 2013*), several substrates showed enrichment within concentric subsets of cells. The most prominent examples of this ordering were pS6-235/236 and pS6-240/244, which shared similar locations of maximal intensity. While the intensity of p4E-BP1 staining in this projection appeared more dispersed and lacked a local region of maximal intensity, cells positive for p4E-BPI1 coincided with pS6-positive cells. Yet compared to pS6, p4E-BP1 staining extended to include progressively larger regions of cells (*Figure 6B*, left column). This pattern is consistent with known mTORC1 substrate sensitivity, such that more sensitive targets (4E-BP1) are phosphorylated at intermediate levels of mTORC1 activity, while less sensitive targets (pS6-S235/236) are not. However, this ordering did not apply to other substrates, including pS6K1, pULK1, and TFEB$_{C/N}$, each of which showed a pattern of intensity distinct from the pS6/p4E-BP1 axis. The maximal points for each of these stains were unique from one another and from the pS6/p4E-BP1 axis, implying that the overall phosphorylation of the mTORC1 substrate network does not occur along a singular axis.

We next examined the relationship of mTORC1 target phosphorylation to protein synthesis by examining OPP labeling. OPP showed an approximate correlation with pULK1, pS6, and p4-EBP1 but was not tightly associated with any one mTORC1 target (*Figure 6B*). We used partial least-squares regression (PLSR) models to examine which measurements (and combinations thereof) best captured variance in OPP labeling. When population averages were used as inputs to the model, 98% of variance could be explained by using the complete set of stains, and 92% using only normal-ized mTORC1 targets – that is, the phosphorylated proteins scaled by their totals (p4E-BP1, pS6-235/236, pS6-240/244) and TFEB$_{C/N}$ (*Figure 6—figure supplement 3*). This result suggests that mTORC1 activity can, in fact, almost completely explain the *average* changes in protein synthesis rate that arise from varying these nutrients (glucose, glutamine, essential amino acids, and insulin). Cell-to-cell variance in protein synthesis, however, is much greater as it includes all effects, not just those directly biased by these average nutrient differences. Nevertheless, all stains together predicted 60% of the cell-to-cell variance in OPP, while the normalized targets of mTORC1 predicted 25% of the variance (*Figure 6C*, left). This modest predictivity is in line with other studies of cell-to-cell heterogeneity; we have previously observed the activity of gene expression signaling pathway providing ~30% prediction of cell-to-cell variance in its canonical targets (*Gillies et al., 2017*). We speculate in this case that because unnormalized measurements reflect differences in absolute abundance, they effectively act as integrated indicators of the protein synthesis rate. Individual mTORC1 targets showed predictivity ranging from <1% to 18%, with pULK1 being the best single predictor (*Figure 6C*, right). Notably, the individual normalized mTORC1 phosphorylation targets (p4E-BP1, pS6-235/236, pS6-240/244, and TFEB$_{C/N}$) each provided <5% prediction individually but afforded much higher predictivity (~25%) when combined. Such cooperativity implies that, as a controller of protein synthesis, mTORC1 activity must be viewed as a network of processes rather than a single linear pathway.

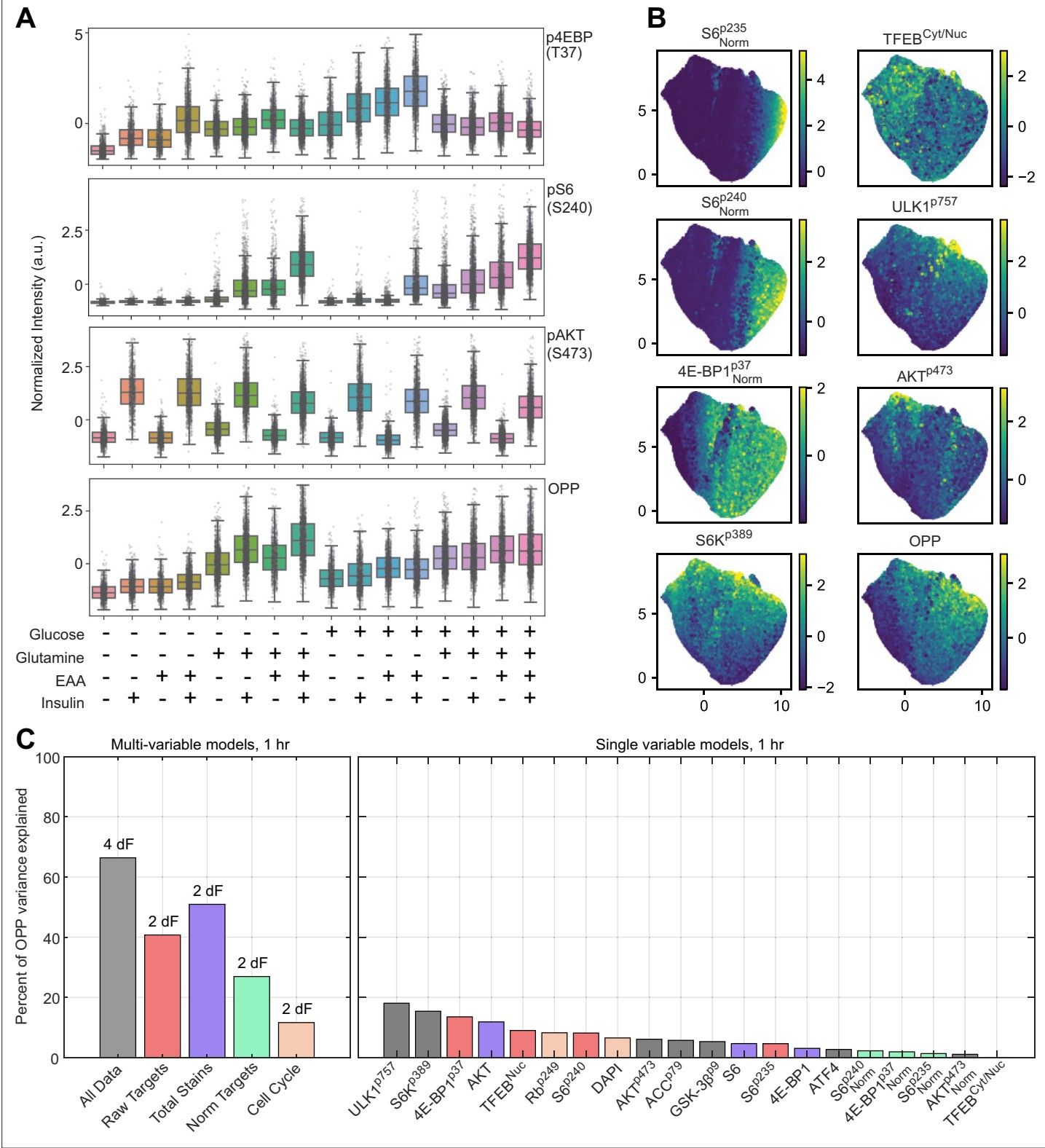

**Figure 6.** Single-cell immunofluorescent analysis of mTORC1 targets and protein expression. (**A**) Normalized distributions of measurements for select targets, under each treatment regimen. Boxplot span 25th–75th percentiles, with a line at the median, and whiskers extend to 5th and 95th percentiles. Individual data are superimposed as small gray dots (*Figure 6—source data 1*). (**B**) UMAP projections of dataset, pseudocolored by intensity of select targets. (**C**) Performance of partial least-squares regression (PLSR) models built to estimate O-propargyl puromycin (OPP) intensity from selected subsets

*Figure 6 continued on next page*

*Figure 6 continued*

of the dataset. Individual targets (right panel) are colored according to the multivariable model that includes them (left panel). Each multivariable model is annotated with the number of degrees of freedom (dF) that was identified by the PLSR.

The online version of this article includes the following source data and figure supplement(s) for figure 6:

**Source data 1.** Cyclic immunofluorescence dataset, w/wo glucose, glutamine, insulin, and amino acids.

**Figure supplement 1.** Sample immunofluorescence images of mTORC1 targets, corresponding to the quantified values in *Figure 6*.

**Figure supplement 2.** Treatment-specific UMAP projection overlays for select targets.

**Figure supplement 3.** Performance and behavior of partial least-squares regression (PLSR) models.

## Discussion

The data presented here provide two central insights that were previously unaddressed experimentally. First, live-cell imaging shows that mTORC1 activity changes within a cell are continuous, both in the sense that mTORC1 responds to various stimuli dose-dependently in continuous gradations rather than large abrupt shifts, and that mTORC1 activity can fluctuate continuously over time, changing on the order of minutes or faster. Second, immunofluorescence analysis demonstrates that different mTORC1 substrates are not co-linearly activated within individual cells. Instead, each responds with different sensitivity, and they are collectively correlated with the cellular protein synthesis rate. Together, these findings elaborate how mTORC1 operates as an analog homeostatic controller for cellular anabolic functions.

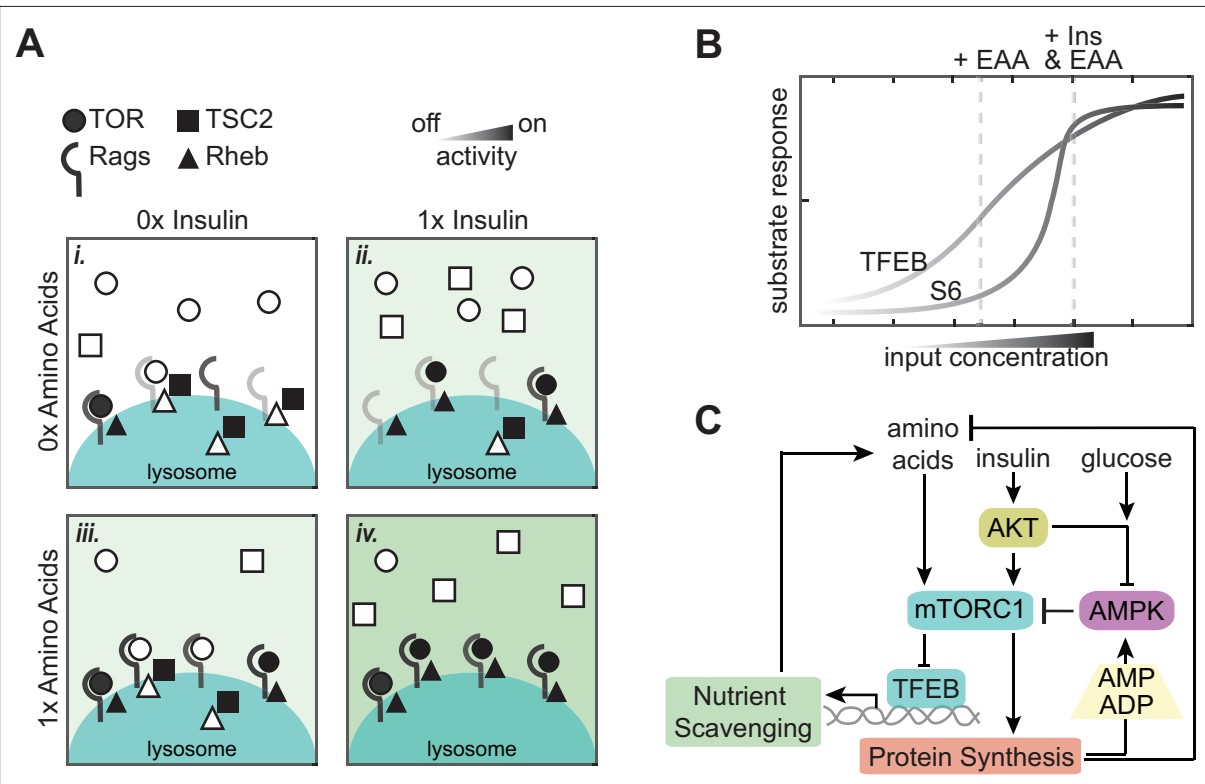

**Figure 7.** A model of continuous integration of nutrient and growth factor status by mTORC1-TFEB signaling. (**A**) Schematic representation of many molecular switches forming an analog integration of nutrient and growth factor abundance by mTORC1 assembly at the lysosome. (**B**) Graph illustrating how differences in mTORC1 substrate sensitivity can change the appearance of mTORC1 integration logic. (**C**) Wiring diagram illustrating potential sources for slow and fast fluctuations in mTORC1 activity. Protein synthesis produces ADP and depletes available amino acids, with potential to activate AMPK and inhibit mTORC1 when nutrients are limited. Transcriptional activity of TFEB promotes nutrient scavenging programs to balance metabolic supply and demand.

## A continuous model for multi-input sensing by mTORC1 at the cellular level

Prior to this work, several observations suggested that mTORC1 activity might be discontinuous. In a number of studies, phosphorylation of mTORC1 substrates appeared to be much greater in response to combined amino acid and growth factor stimuli than in response to either stimulus alone (*Carroll et al., 2016*; *Menon et al., 2014*). Furthermore, immunofluorescence for pS6 often shows stark cell-to-cell variation in intensity (*Figure 6A and B* and *Figure 6—figure supplement 1*; *Hirashita et al., 2021*; *Rüschoff et al., 2022*), which can imply all-or-none regulation. However, regardless of whether dose curves of amino acids or growth factors are presented individually or in combination, our live-cell data indicate that mTORC1 activity responds continuously, ruling out fundamentally 'digital' or 'switch-like' mechanisms where the response is focused on a sharp threshold. Our single-treatment experiments (*Figure 3B–E*) support that mTORC1 presents an analog response, increasing proportionally with the addition of either amino acids or insulin.

While our findings might appear to conflict with the molecular AND gate model for mTORC1 regulation by Rheb and Rag, the difference can be reconciled by supposing that the basal levels of active Rag heterodimers and GTP-bound Rheb are low, but not zero (*Figure 7A*, panel i). Given this scenario, increasing *either* the activated Rag-complex binding sites or reducing the number of TSC2 complexes on the lysosome would be sufficient to increase the number of active mTORC1 complexes (*Figure 7A*, panels ii and iii). However, both stimuli would be needed to reach maximal activity. In other words, activation of each individual mTORC1 behaves as a molecular AND gate requiring engagement with both an active Rag and active Rheb (*Saxton and Sabatini, 2017*), but the number of such productive complexes within the cell would be widely variable, allowing the total cellular activity of mTORC1 to range continuously as a sort of cellular rheostat.

This model underscores the importance of clarifying the relationship between molecular mechanisms (i.e. how individual molecular complexes function) and systems-level behaviors (i.e. how the many copies of the complex behave collectively within the cell). The boolean logic of individual mTORC1 complexes does not imply strict on/off responses at the cellular level, but there are important cellular implications for this molecular behavior. Our results using sequential stimuli (*Figure 4*) suggest that the spatial organization of mTORC1 activation creates a motif that prevents any single stimulus from exerting a saturating effect on mTORC1. For example, even if excess insulin saturates the insulin receptor or PI3K/AKT pathway, mTORC1 activity can remain sensitive to changes in amino acids by the independent interaction with Rag. If both growth factor and amino acid stimuli converged on the same molecular mechanism – effectively creating a molecular OR gate – it would be possible for one stimulus alone to saturate mTORC1, overriding regulation by the other stimulus. Thus, while the statistics of having many copies of each regulatory factor preclude a binary response, the molecular AND gate motif may yet be critical in protecting the ability of mTORC1 to remain responsive to multiple inputs.

## Differences in additive behavior among mTORC1 substrates

Our data also show that the appearance of synergistic mTORC1 activation depends on which substrate is used to track its activity, and whether phosphorylation is assessed at a single time point or over time. Notably, phosphorylation of S6K or its substrate S6 have been used in most studies that have observed a strict requirement for both amino acids and growth factors (*Carroll et al., 2016*; *Menon et al., 2014*). However, we find that among the substrates we tested, pS6 is the most atypical at the single-cell level, with an unusually high degree of cell-to-cell heterogeneity. A broader view of multiple substrates suggests that the activity of mTORC1 itself is graded, while its substrates vary in their nonlinear response to this activity. Differences between mTORC1 substrates have been explored previously, revealing that the affinity of mTORC1 for different phosphorylation sites predicts their responses to pharmacological inhibition of mTOR (*Kang et al., 2013*). These differences suggest a simple model for the variation we observe between substrates, based on typical sigmoidal dose–response curves (*Figure 7B*). A super-additive effect would be expected for weaker substrates such as S6K because each individual stimulus would lie within the threshold region of the response curve, and the response to the combined stimuli would be amplified by the upward concavity of the sigmoid. For more sensitive substrates, both individual stimuli and their combination would lie within the approximately linear region of the response curve, resulting in an additive effect. Based on measurements of

substrate affinity (*Kang et al., 2013*), this model could explain the differences in additivity between 4E-BP1 or ULK1 and S6.

Additionally, it is possible that each of the nodes in the network is co-regulated by additional signals, which could explain the divergent behavior of pS6. For example, Ser 235/236 of S6 can be phosphorylated both by p70 S6K and by p90 RSK (*Pende et al., 2004*), which is typically activated in response to MAPK signaling (though MAPK/RSK activity is canonically unexpected in the conditions used for this immunofluorescence study). In the case of TFEB, the response to mTORC1 is modulated by a cluster of phosphorylation sites that confer regulation by GSK3, mTORC2, and PKC (*Figure 2B*; *Li et al., 2018*; *Napolitano et al., 2018*). This cluster operates by regulating TFEB export from the nucleus by Crm1, in parallel to the regulation of cytoplasmic retention by 14-3-3 binding to Ser-211, allowing TFEB itself to integrate multiple inputs downstream of mTORC1. Notably, however, we observe that mutation of the GSK3β-linked sites, or withdrawal of glucose, does not abrogate the ability of TFEB to respond incrementally to multiple stimuli (*Figure 4—figure supplement 1*). We infer that GSK3β-mediated phosphorylation of TFEB is not strictly essential for integrating multiple stimuli, but rather serves to amplify the responsiveness of TFEB to mTORC1. An interpretation of previous data implied that GSK3β regulation is strictly required for TFEB nuclear export *Li et al., 2018*; however, we argue that the two studies are reconcilable, again by interpreting all of the available data quantitatively. TFEB exchanges continuously between the nucleus and cytoplasm (*Li et al., 2018*; *Napolitano et al., 2018*), and nuclear export of Ser-142 or Ser-138 mutants is reduced partially (by 75%) rather than fully eliminated (*Napolitano et al., 2018*). Therefore, TFEB localization likely reflects a balance between the Ser-211/14-3-3 and Ser-142/Crm-1 mechanisms rather than a strict reliance on either one. In this view, GSK3β can be thought of as an amplifier for TFEB responsiveness, which parallels the finding that GSK3β enhances the activity of mTORC1 on other substrates, including S6K1 and 4E-BP1 (*Shin et al., 2014*; *Shin et al., 2011*), consistent with a general role for GSK3β as a signaling amplifier of previously primed motifs (*Hermida et al., 2017*).

## Operational analysis of mTORC1 and implications for physiology

Physiologically, mTORC1 activity is required for embryonic development, but its inactivation is also essential. Mice deficient for TSC1 or TSC2 cannot suppress mTORC1 upon growth factor withdrawal and die during embryogenesis (*Kobayashi et al., 1999*; *Kobayashi et al., 2001*); *in vitro* studies of this model suggest that cell death results from a failure to limit energy expenditure (*Choo et al., 2010*). In mice engineered to express a GTPase-deficient (GTP-locked) RagA, mTORC1 remains active upon nutrient starvation, and the resulting death of neonatal mice is linked to a failure to induce autophagy (*Efeyan et al., 2013*). Our observation that TFEB-TR$_{C/N}$ correlates tightly with fluctuations in AMPK activity on the scale of minutes is consistent with this concept, indicating that mTORC1 plays a key role in tracking energetic challenges that might occur under physiological conditions. Further nuance is added by the finding that, under varying nutrient conditions, mTORC1 substrate phosphorylation can predict up to 25% of single-cell differences in protein synthesis. This fraction indicates significant control over cellular energy usage, sufficient to be physiologically essential during development, but it also indicates that mTORC1 alone does not dictate synthesis rates. On reflection, this is to be expected under the model of mTORC1 as a *permissive* controller, where it enforces soft limits on protein synthesis at any given time, but not does not force synthesis in the absence of other drivers of expression.

In summary, the multi-reporter measurements in this study help to conceptualize cell-intrinsic homeostasis mechanisms not as separate alarm-like pathways, but as a well-connected system of continuous controllers, akin to whole-body feedback systems such as glucose or electrolyte regulation (*Figure 7C*). To date, a major impediment to building on this more holistic concept has been the technical limitation on making detailed dynamic measurements at such small scales. Fortunately, a growing body of single-cell techniques is beginning to enable such characterization and empower the application of quantitative models. An immediate and critical challenge to address is further multiplexing single-cell techniques to expand the repertoire of live-cell datasets that link multiple signals with metabolic changes. A key future step will be to build formal quantitative models for how mTOR homeostatic control operates and interacts with other cellular systems. The strong coupling of signals (i.e. AMPK to mTORC1), but incomplete control over outputs (i.e. protein synthesis) demonstrates the need for a global view of the metabolic signaling network, especially in terms of interacting factors.

## Limitations of the study

Our results are based largely on experiments conducted in a single, genetically stable, immortalized cell line, and thus the relevance of our conclusions for most other cell types or for any particular tissue within the body is not known. All experiments were performed *in vitro* in cell culture, where variables such as oxygen levels, which are known to affect mTORC1 activity, were significantly outside the range that cells would encounter *in vivo*. This caveat also extends to the concentrations of various macro- and micronutrients (sugars, amino acids, vitamins), which in many cases were significantly different than those found in human tissues. Our findings should thus be interpreted as characterizing a set of behaviors that are possible for the mTORC1-TFEB network within a certain context. These results should be further tested under relevant conditions before concluding that they represent the behavior of the network in other cell types.

# Materials and methods

### Reporter construction

Plasmids encoding full-length human TFEB (FL-TFEB) were received as a gift from the Zoncu lab. The series of TFEB reporters, including FL-TFEB, were constructed in the pLJM1 lentiviral vector, and the coding region of TFEB was modified, as shown in *Figure 2*. The C-termini of these modified TFEB constructs were fused with either mVenus or mCardinal for visualization. TFEB-TR was constructed by truncating FL-TFEB to amino acids 1–237. 3xSA-TR was constructed by site-directed mutagenesis of amino acid residues (S134A, S138A, and S142A) on the TFEB-TR construct. Similarly, 5xSA-TR was constructed by site-directed mutagenesis of amino acid residues (S122A, S134A, S138A, S142A, and S211A) on the TFEB-TR construct. TOP-H2B-YFP-DD was received as a gift from the Meyer lab and is constructed and used as described in *Han et al., 2014*. FOXO-TR was constructed as described in *Hung et al., 2017* with the fluorophore changed to mOrange for spectral compatibility and expressed using retroviral transduction. AMPKAR2 construction is as described in *Hung et al., 2017* and was expressed in the pPBJ, piggyBAC transposase delivery system (*Yusa et al., 2011*).

### Cell line creation

Retroviral or PiggyBac systems were used to generate stably expressing reporter cell lines. Following transfection, cells were selected using puromycin (1–2 µg/mL), G418 (200 µg/mL), or geneticin (300 µg/mL). To reduce variability in reporter expression, clonal cell lines were established using limited dilution cloning. In this study, data for each reporter is representative of behaviors that were consistent across at least three single-cell clones. Reporter cell lines were confirmed mycoplasma-negative through periodic monitoring by PCR and validation with ATCC testing of selected lines.

### Cell culture

MCF10A cells clone 5E (*Janes et al., 2010*) were routinely maintained in 'DMEM/F12 growth media' (see Media composition table) as described in *Debnath et al., 2003*. Wild-type MEF cells (a gift from Arnim Pause) were maintained in 'DMEM MEF growth media.' For live-cell time-lapse microscopy experiments, we used custom formulations, termed 'Imaging base-DMEM/F12,' which consists of DMEM/F12 lacking glucose, glutamine, riboflavin, folic acid, and phenol red (Life Technologies or UC Davis Veterinary Medicine Biological Media Service) to avoid fluorescence background. 'Imaging base-DMEM/F12-noAA' is the equivalent, but also excluding amino acids. All experiments that did not involve amino acid perturbation were performed in 'Imaging medium' (see Media table). 'Imaging medium – noAA' was used in experiments that involves amino acid perturbation (see Media table). 'MEF imaging medium – noAA' was used for experiments with MEF cells.

Before imaging, cells were washed twice with their respective imaging media and then cultured in imaging experiment media at least 2 hr prior to imaging. In experiments that involved amino acid perturbation, cells were pre-incubated with 'Imaging medium – noAA' for at least 6 hours prior to imaging to limit residual extra amino acids from the routine cell culture media.

### Immunofluorescence microscopy

For all imaging experiments, cells were seeded in glass-bottom 96-well plates (Cellvis P96-1.5H-N, Mountain View, CA) 1 d prior to imaging. To increase the cell-to-media ratio, cells were plated in the

center of the well by spotting 9000 cells on 3 uL of type I collagen (Gibco A10483-01). Following the treatment regimen, cells were fixed in 4% paraformaldehyde solution for 15 min, and then permeabilized with 100% methanol for 15 min.

For immunostaining, cells were washed in PBS-T (0.1% Tween-20 in PBS) twice, blocked for 1 hr with Odyssey Blocking Buffer (Li-Cor, Lincoln, NE). A sequential staining protocol (*Gut et al., 2018*; *Lin et al., 2016*) was used in which the sample was blocked (200 mM ammonium chloride, 300 mM maleimide, 2% bovine serum albumin in PBS), and incubated overnight with primary antibodies at 4°C, followed by secondary antibodies and Hoechst-33342 (Life Technologies, H3570, diluted at 1:1000 in PBS). After image acquisition, antibodies were eluted using 0.5 M glycine, 3 M urea, 3 M guanidinium chloride, and 70 mM, and the sample was subjected to additional rounds of staining and imaging to acquire multiplexed images of the same cells. Images were collected with a Nikon (Tokyo, Japan) 20/0.75 NA Plan Apo objective on a Nikon Eclipse Ti inverted microscope, equipped with a Lumencor SOLA or Lumencor SPECTRA X light engine.

## Live-cell fluorescence microscopy

For live-cell microscopy experiments, cells were cultured and seeded as described in previous sections. Time-lapse wide-field microscopy was performed as detailed in *Tan and Huang, 2017*, with cells maintained in 95% air and 5% $CO_2$ at 37°C in an environmental chamber. Images were collected with a Nikon (Tokyo, Japan) 20x/0.75 NA Plan Apo objective on a Nikon Eclipse Ti inverted microscope, equipped with a Lumencor SOLA or Lumencor SPECTRA X light engine. Fluorescence filters used in the experiment are DAPI (custom ET395/25x – ET460/50m – T425lpxr, Chroma), CFP (49001, Chroma), YFP (49003, Chroma), Cherry (41043, Chroma), and Cy5 (49006, Chroma). Images were acquired using Andor Zyla 5.5 sCMOS camera every 6–7 min at 2 × 2 binning. Exposure times for each channel were 25–50 ms for DAPI; 150–250 ms for CFP; 150–250 ms for YFP; 300–500 ms for Cherry; and 300–500 ms for Cy5.

## Image processing and time-series data selection criteria

Live-cell microscopy images were processed using the custom MATLAB procedure as described in our previous work (*Tan and Huang, 2017*). In this procedure, the nuclear and cytoplasmic region of individual cells are segmented, and the fluorescent intensity of pixels within the masked regions are averaged. For this study, the segmentation protocol was modified to restrict the cytoplasmic masks to the region within 5 pixels from the nuclear border, and the nuclear masks were eroded 2 pixels away from the cytoplasmic border. For all imaging experiments shown, a minimum of 100 cells were imaged and tracked for each condition.

The single-cell time-series data were further filtered, requiring minimum fluorescent intensity more than two times background intensity, trace length of at least 8 hr, and no more than three contiguous missing datapoints. Missing datapoints in time series were linearly interpolated to fill the gap.

## Reporter activity analysis

For all translocation reporters (FL-TFEB, TFEB-TR, 3xSA-TR,5xSA-TR, and FOXO-TR reporter), the activities of the kinases are described by the ratio of average cytosolic fluorescent intensity divided by average nuclear fluorescent intensity.

For the AMPKAR2 reporter, AMPKAR2 phosphorylation status was calculated using the protocol described in *Kosaisawe et al., 2021*. Briefly, linearized AMPKAR2 FRET efficiency was calculated as shown in our previous work (*Gillies et al., 2020*). Then AMPKAR2 phosphorylation status was estimated based on this equation

$$AMPKAR2^{PHOS} = 2.74 \left[ AMPKAR2_{FRET\ ratio} \right] - 0.59$$

For the TOP-H2B-YFP-DD reporter, the rate of intensity gain over time was measured after addition of trimethoprim (TMP), before and after a stimulus. The ratio of the rate after to that before the stimulus is used to represent the relative translation activity (RTA) induced by the stimulus.

## Time-series volatility analysis

For all volatility analysis of time-series data, we used a time window between 2 and 6 hr after the final perturbation. We estimated that in this time window the kinase activities of interest have approached a quasi-steady state. Within that time window, data from every other time step were then used to calculate CV, which we used as metric of volatility. Alternating time steps were used to mitigate non-biological noise, such as imaging and segmentation noise.

## Cell trace examples

The displayed time series were chosen by random number generation in MATLAB with a threshold for minimum tracking time to eliminate cells in which recording was terminated prematurely due to failure of the tracking algorithm. The chosen tracks were manually verified to be representative of successfully tracked cells and consistent with the overall range of cell behaviors. Cell recordings determined by manual inspection to have poor tracking or quantification accuracy were discarded.

## Linear fitting and statistical tests

Linear regression analysis was performed using a MATLAB implementation of the Theil-Sen estimator, which is relatively insensitive to outliers and robust to heteroskedastic data. Unless otherwise indicated, each statistical comparison was made by $t$-test with unequal variances. A p-value of 0.05 is considered as significant for all hypothesis testing.

## Inhibitors

| Name | Solvent | Vendor | Catalog number |
|---|---|---|---|
| Torin2 | DMSO | Selleckchem | S2817 |
| Rapamycin | DMSO | Selleckchem | S1039 |
| Essential amino acid | Water | Gibco | 11130051 |
| MK2206 | DMSO | Selleckchem | S1078 |
| CHIR99021 | DMSO | Tocris | 4423 |
| A769662 | DMSO | Tocris | 3336 |
| Glucose | Water | Fisher Scientific | D16-500 |
| L-Glutamine | | Thermo Fisher | A2916801 |
| Oligomycin A | DMSO | Sigma | 75351 |

## Primary antibodies

| Name | Target region | Dilution | Vendor | Catalog number |
|---|---|---|---|---|
| TFEB total | ca. Gly412 | 1:600 | CST | 37785 |
| TFEB total M | | 1:100 | MyBioSource | 120432 |
| 4EBP-1 p-37/46 | Thr37/46 | 1:200 | CST | 2855 |
| 4EBP-1 total | ca. Ser112 | 1:200 | CST | 34470 |
| ULK1 p-757 | Ser757 | 1:800 | CST | 14202 |
| S6 p-235/236 | Ser235/236 | 1:200 | CST | 4858 |
| S6 p-240/244 | Ser240/244 | 1:500 | CST | 5364 |
| S6 total | Full length | 1:200 | CST | 5548 |
| S6K p-389 | Thr389 | 1:200 | Invitrogen | 710095 |
| ACC p-79 | Ser79 | 1:400 | CST | 11818 |

*Continued on next page*

*Continued*

| Name | Target region | Dilution | Vendor | Catalog number |
|------|---------------|----------|--------|----------------|
| AKT p-473 | Ser473 | 1:100 | CST | 4071 |
| AKT total | C-terminus | 1:100 | CST | 5186 |
| GSK p-9 | Ser9 | 1:200 | CST | 14026 |
| GSK-3b total | C-terminus | 1:250 | CST | 9832 |
| Rb p-249/252 | Ser249/Thr252 | 1:400 | SCBT | 377528 |

## Secondary antibodies

| Name | Conjugate | Dilution | Vendor | Catalog number |
|------|-----------|----------|--------|----------------|
| Goat anti-rabbit 647 | Alexa Fluor 647 | 1:250 | Life Technologies | A-21245 |
| Goat anti-rabbit 555 | Alexa Fluor 555 | 1:250 | Life Technologies | A-21428 |
| Goat anti-rabbit 488 | Alexa Fluor 488 | 1:250 | Life Technologies | A-11034 |
| Goat anti-mouse 488 | Alexa Fluor 488 | 1:250 | Life Technologies | A-11001 |

## Cell lines

| Number | Name | Source | Identifier |
|--------|------|--------|------------|
| 1 | MCF10A | *Hung et al., 2017* | RRID:CVCL_0598 |
| 2 | MCF10A_FL-TFEB-TR | This report | |
| 3 | MCF10A_TFEB-TR | This report | |
| 4 | MCF10A_3XSA-TFEB-TR | This report | |
| 5 | MCF10A_5XSA-TFEB-TR | This report | |
| 6 | MCF10A_TFEB-TR_FOXO-TR_AMPKAR2 | This report | |
| 7 | WT-MEF_TFEB-TR | This report | |

## Media composition

**DMEM/F12 growth medium**

| Component | Stock concentration | Volume | Unit | Vendor | Catalog number | Final concentration |
|-----------|---------------------|--------|------|--------|----------------|---------------------|
| DMEM/F2 | - | 500 | mL | Gibco | 11320-033 | - |
| Horse serum | - | 25 | mL | Invitrogen | 16050-122 | 5% |
| EGF | 100 µg/mL | 100 | uL | Peprotech | AF-100-15 | 20 ng/mL |
| Hydrocortisone | 1 mg/mL | 250 | uL | Sigma | H0888 | 0.5 mg/mL |
| Cholera toxin | 1 mg/mL | 50 | uL | Sigma | C8052 | 100 ng/mL |
| Insulin | 10 mg/mL | 500 | uL | Sigma | I9278 | 10 µg/mL |
| DMEM MEF growth medium | | | | | | |
| DMEM | - | 500 | mL | Gibco | 11965-092 | - |
| FBS | - | 50 | mL | GeminiBio | 100-106 | 10% |
| Penicillin-streptomycin | 10,000 U/mL | 5 | mL | Gibco | 15140122 | 100 U/mL |

*Continued on next page*

*Continued*

### DMEM/F12 growth medium

| Component | Stock concentration | Volume | Unit | Vendor | Catalog number | Final concentration |
|---|---|---|---|---|---|---|
| Imaging medium | | | | | | |
| Imaging base- modified DMEM/F12 | - | 500 | mL | Gibco or UC Davis Veterinary Medicine Biological Media Service | Custom recipe produced at UC Davis, equivalent to Gibco 11320-033 lacking glucose, glutamine, pyruvate, riboflavin, folic acid, and phenol red | |
| BSA | 2% w/v | 25 | mL | Invitrogen | 16050-122 | 0.1% w/v |
| Hydrocortisone | 1 mg/mL | 250 | uL | Sigma | H0888 | 0.5 mg/mL |
| Cholera toxin | 1 mg/mL | 50 | uL | Sigma | C8052 | 100 ng/mL |
| Penicillin-streptomycin | 10,000 U/mL | 5 | mL | Gibco | 15140122 | 100 U/mL |
| L-Glutamine | 200 mM | 6.25 | mL | Gibco | A2916801 | 2.5 mM |
| Folic acid | 10 mg/mL | 212 | uL | Sigma | F8758 | 4.24 µg/mL |
| Riboflavin | 10 mg/mL | 16.25 | uL | Sigma | R9504 | 0.352 µg/mL |
| Imaging medium – noAA | | | | | | |
| Imaging base-DMEM/F12-noAA | - | 500 | mL | Gibco | Custom; equivalent to Gibco 11320-033 lacking glucose, glutamine, amino acids, pyruvate, riboflavin, folic acid, and phenol red | |
| BSA | 2% w/v | 25 | mL | Invitrogen | 16050-122 | 0.1% w/v |
| Hydrocortisone | 1 mg/mL | 250 | uL | Sigma | H0888 | 0.5 mg/mL |
| Cholera toxin | 1 mg/mL | 50 | uL | Sigma | C8052 | 100 ng/mL |
| Penicillin-streptomycin | 10,000 U/mL | 5 | mL | Gibco | 15140122 | 100 U/mL |
| L-Glutamine | 200 mM | 6.25 | mL | Gibco | A2916801 | 2.5 mM |
| Folic acid | 10 mg/mL | 212 | uL | Sigma | F8758 | 4.24 µg/mL |
| Riboflavin | 10 mg/mL | 16.25 | uL | Sigma | R9504 | 0.352 µg/mL |
| MEF Iimaging medium -– noAA | | | | | | |
| Imaging base-DMEM/F12-noAA | - | 500 | mL | Gibco | Custom; equivalent to Gibco 11320-033 lacking glucose, glutamine, amino acids, pyruvate, riboflavin, folic acid, and phenol red | |
| BSA | 2% w/v | 25 | mL | Invitrogen | 16050-122 | 0.1% w/v |

## Acknowledgements

Funding for this work was provided by the National Institute of General Medical Sciences (1R01GM115650 and R35). Flow-cytometry services were supported by the UC Davis Comprehensive Cancer Center Support Grant (CCSG) awarded by the National Cancer Institute (NCI P30CA093373). We thank Roberto Zoncu for providing plasmids encoding TFEB and for helpful discussions, Arnim Pause for providing mouse embryonic fibroblasts, and Tobias Meyer for generously providing the TOP-H2B-YFP-DD reporter.

## Additional information

### Competing interests

John G Albeck: has received research grants from Kirin Corporation. The other authors declare that no competing interests exist.

### Funding

| Funder | Grant reference number | Author |
|---|---|---|
| National Institute of General Medical Sciences | R35GM139621 | John G Albeck |
| National Institute of General Medical Sciences | R01GM115650 | John G Albeck |
| National Science Foundation | 2136040 | John G Albeck |
| National Heart, Lung, and Blood Institute | T32HL007013 | Nicholaus DeCuzzi |
| National Institute of General Medical Sciences | F31GM120937 | Breanne Sparta |

The funders had no role in study design, data collection and interpretation, or the decision to submit the work for publication.

### Author contributions

Breanne Sparta, Conceptualization, Resources, Data curation, Formal analysis, Investigation, Visualization, Methodology, Writing – original draft, Writing – review and editing; Nont Kosaisawe, Conceptualization, Data curation, Formal analysis, Validation, Investigation, Visualization, Methodology; Michael Pargett, Conceptualization, Resources, Software, Formal analysis, Supervision, Validation, Visualization, Methodology, Writing – original draft, Writing – review and editing; Madhura Patankar, Nicholaus DeCuzzi, Formal analysis, Visualization; John G Albeck, Conceptualization, Supervision, Funding acquisition, Methodology, Writing – original draft, Project administration, Writing – review and editing

### Author ORCIDs

Breanne Sparta http://orcid.org/0000-0002-8255-1409
Nont Kosaisawe http://orcid.org/0000-0002-4710-1082
Michael Pargett http://orcid.org/0000-0002-1403-7408
John G Albeck https://orcid.org/0000-0003-2688-8653

### Decision letter and Author response

Decision letter https://doi.org/10.7554/eLife.74903.sa1
Author response https://doi.org/10.7554/eLife.74903.sa2

## Additional files

### Supplementary files
• Transparent reporting form

### Data availability

Source data files contain the per-cell fluorescence intensity data, extracted from microscope image data, that were used to generate each figure in the paper.

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
