## [Editor Report]

In this study, Sparta et al., performed time-resolved monitoring of the mechanistic target of rapamycin complex 1 (mTORC1) response to stimuli including glucose, amino acids, and insulin at the single-cell resolution. The authors found that mTORC1 activation is graded, which supports the model of continuous adjustment of mTORC1 signaling to changes in the cellular environment rather than an "on/off" mode of mTORC1 function. Considering the pivotal role of mTORC1 in sensing a plethora of signals to orchestrate metabolic programs impacting cell fate and organismal physiology, it was thought that this study will be of interest across a broad spectrum of biomedical disciplines.

---

## [Decision Letter]

**Decision letter after peer review:**

Thank you for submitting your article "Continuous sensing of nutrients and growth factors by the mTORC1-TFEB axis" for consideration by *eLife*. Your article has been reviewed by 3 peer reviewers, including Ivan Topisirovic as Reviewing Editor and Reviewer #1, and the evaluation has been overseen by Philip Cole as the Senior Editor.

Essential revisions:

1) It was thought that presented data support the present view of mTORC1 regulation rather than establishing a previously unappreciated model of stimuli-sensing by mTORC1. As illustrated in the individual reviews, the claim that the present view in the field is that mTORC1 functions analogous to an AND gate was found to be somewhat inaccurate. To this end, it was thought that the authors should focus on highlighting the power of their quantitative approaches that allow time-resolved measurements of mTORC1 activity at the single cell resolution that drew substantial enthusiasm from the reviewers, while deemphasizing the claim that their study challenges current paradigms of mTORC1 regulation. Moreover, it was thought that the focus should also be shifted to what were found to be novel findings (i.e., results presented in figures 5 and 6).

2) Some important controls were deemed to be missing. For instance, it should be verified whether manipulations in glucose, amino acid or insulin levels affect expression of TFEB reporters. In addition, the activity of PI3K/AKT signaling should be examined across the conditions as it was thought that this may be crucial to determine the extent of the effects of starvation and amino acid stimulation on mTORC1 signaling in the absence of insulin. Moreover, pharmacological and/or genetic approaches to verify whether employed stimuli are still capable to activate mTORC1 signaling when PI3K/AKT signaling is debilitated appear to be warranted. Finally, monitoring Ser235/236 phosphorylation of rpS6 may not to reflect alterations in signaling via the mTORC1/S6K axis.

3) Some concerns were raised regarding the differences in cell density between baseline and treatment conditions in the examples of original imaging data that have been provided. Based on this, it was thought that more original data should be included to support the quantification plots.

4) In some instances, the information provided in the methodology section lacked sufficient detail.

5) The authors should acknowledge and/or experimentally address limitations of their study pertinent to relying on a single cell line and excluding physiological variables that are known to affect mTORC1 activity (e.g., oxygen levels).

*Reviewer #1 (Recommendations for the authors):*

– The authors propose that the prevalent model of mTORC1 regulation is that analog to AND gate type of digital logic gate, whereby it is assumed mTORC1 is only active in the presence of both amino acids and growth factors under optimal cellular energy conditions. This is likely to be my subjective interpretation, but I do not think that such AND gate model of mTORC1 regulation is strongly supported by the literature. In contrary, I think that the prevailing model of mTORC1 regulation is that of a rheostat that is akin to the one proposed by the authors. With this in mind, my suggestion to the authors is not to claim the novelty of the model of mTORC1 regulation, but rather their ability to support the existing model by quantifying the effects of various stimuli on mTORC1 activity at the single-cell level, which at least I found to be quite exciting and warranted.

– The major limitation of the study is that it based on a single cell line. This renders generality of the authors' conclusions somewhat questionable and should be either accompanied by testing other cell lines or commented as a limitation of the study in the manuscript. Along the same lines, the authors should consider toning down claims of how relative their findings are physiologically as although they study the effects of glucose, amino acids, and insulin on MCF10A cells, it is unlikely that their model reflects the complexity of the physiological environments which comprise fluctuations in oxygen and other factors.

– Methods section regarding the exact conditions of starvation/stimulation should perhaps be further clarified. In addition, the authors should consider providing evidence that 6h starvation of MCF10A cells is sufficient to shut down PI3K/AKT signaling. Would mTORC1 be activated by amino acids in the presence of e.g., AKT inhibitor? It was also thought that it would be important to monitor potential effects of manipulations in nutrient and/or insulin on reporter expression.

– Phosphorylation of S235/236 on rpS6 can be independent of the mTORC1/S6K axis and its regulation is unaltered in cells lacking S6K1 and 2 (PMID: 15060135). Ergo, using p-S235/236 on rpS6 as a readout of mTORC1 activity may be problematic.

*Reviewer #2 (Recommendations for the authors):*

1. Does the amount of nutrients/growth signals added affect total expression of TFEB fusion proteins and mutants? Is this taken into consideration in the calculation of C/N particularly during prolonged periods?

2. The findings in Figures4 and 5 that TFEB-C/N responds to amino acids despite absence of insulin should be verified for the presence of insulin/growth factor signaling. According to the Methods, cells were starved for 6 hr. There could be residual insulin/PI3K signaling, thus allowing TFEB response upon amino acid addition. They should examine PI3K/Akt signaling. Would this aa response still occur upon inhibition of PI3K or Akt? What happens when starvation is prolonged to 12-16 hr before re-addition of nutrients?

*Reviewer #3 (Recommendations for the authors):*

This is an interesting manuscript from Sparta and colleagues that investigates dynamics of MTOR and TFEB signalling.

While this reviewer appreciates the elegance and power of the experimental system, and the substantial amount of data in the manuscript, we did not find that the data provide enough clear insight into the mechanisms of MTOR regulation or signalling to justify publication in *eLife*. For publication as a full-length paper, it would seem the narrative would need to be re-structured to highlight how findings illustrate novel insights with relevance that can be appreciated by a broad audience. The paper should be revised to focus on a smaller number of specific biological questions that could be addressed by the single cell signalling data.

For example, the authors were able to demonstrate that the truncated TFEB-TR reporter behaves similarly to full length, but that a 3xSA-TFEB-TR lacking GSK3beta modification sites was poorly responsive, but these are not significant enough advances for the field.

The authors aimed to use cell based reporters to investigate digital vs gradual modes of MTOR signalling, an interesting concept for signalling systems biology. Data in Figure 4 support the notion of incremental responses to amino acids and also to insulin signalling. The study proceeds with investigation of multiple signalling inputs, developing highly granular analysis of sequential additions. The biological mechanistic insight from these experiments remains unclear. Additional information may be gained from analysis of different combinations of simultaneous stimulation eg AA+insulin, as explored initially in the manuscript. Linking further with the initial part of the paper, it is interesting that TFEB responses differed with other MTOR downstream signalling, most notably with P-S6, but also with P-ULK1-S757. It would be interesting if further mechanistic insight could be demonstrated by modulation of responses by gene silencing of essential components of AA sensing, insulin signalling or TFEB signalling pathway. In addition, this paper would be greatly strengthened if some of the cell-based imaging readouts can be complemented by cell population immuno-blot analysis for key experiments.

---

## [Author Response]

Essential revisions:1) It was thought that presented data support the present view of mTORC1 regulation rather than establishing a previously unappreciated model of stimuli-sensing by mTORC1. As illustrated in the individual reviews, the claim that the present view in the field is that mTORC1 functions analogous to an AND gate was found to be somewhat inaccurate. To this end, it was thought that the authors should focus on highlighting the power of their quantitative approaches that allow time-resolved measurements of mTORC1 activity at the single cell resolution that drew substantial enthusiasm from the reviewers, while deemphasizing the claim that their study challenges current paradigms of mTORC1 regulation. Moreover, it was thought that the focus should also be shifted to what were found to be novel findings (i.e., results presented in figures 5 and 6).

We agree that this shift in perspective is helpful, and we have refocused the work on the quantitative gains of our live-cell measurements and sequential immunofluorescent stains. We also now provide more formal quantitative modeling to address how mTORC1 activity relates quantitatively to protein synthesis rates. The manuscript text has been edited and reorganized according to this newly refined focus. To provide a complete discussion of the existing literature, we have kept a short introduction to the AND gate concept, but we no longer claim that it is the predominant view of the field or that the novelty of our study lies in the refutation of this concept.

2) Some important controls were deemed to be missing. For instance, it should be verified whether manipulations in glucose, amino acid or insulin levels affect expression of TFEB reporters.

A control for total expression of TFEB reporter (TFEB C+N) in response to amino acid stimulus is now provided as Figure 1—figure supplement 1F.

In addition, the activity of PI3K/AKT signaling should be examined across the conditions as it was thought that this may be crucial to determine the extent of the effects of starvation and amino acid stimulation on mTORC1 signaling in the absence of insulin.

AKT activity in live cells is reported now in Figure 4, across serial stimulation experiments including:

starvation, glucose/glutamine re-feeding, amino acid re-feeding, and insulin treatments.

Moreover, pharmacological and/or genetic approaches to verify whether employed stimuli are still capable to activate mTORC1 signaling when PI3K/AKT signaling is debilitated appear to be warranted.

Pharmacological inhibition of AKT is reported in Figure 3—figure supplement 1C (using MK2206). In this panel, cells received full amino acids; note that upon MK-2206 treatment, TFEB-TR localization returns to the C/N level for amino-acid stimulated cells (about 1.0), not to the amino acid starved baseline level (about 0.8) as observed in Figure 3B. If amino acids were unable to activate mTORC1 signaling in the presence of the inhibitor, we would expect the TFEB C/N ratio to drop to the starved baseline of 0.8.

Finally, monitoring Ser235/236 phosphorylation of rpS6 may not to reflect alterations in signaling via the mTORC1/S6K axis.

We agree that the interpretation of data from this phospho-epitope needs to take into account the fact that it can be phosphorylated by both p70 S6K and p90 RSK. This convergence is one mode by which alternate regulation can come into play. In the revised manuscript, measurements of substrate phosphorylation are now provided all together in Figure 6, so that the signaling targets may be evaluated in a broader context. We have also now directly commented on this pathway convergence in the results and discussion.

3) Some concerns were raised regarding the differences in cell density between baseline and treatment conditions in the examples of original imaging data that have been provided. Based on this, it was thought that more original data should be included to support the quantification plots.

Example images are now found in Figure 1 and Figure 6—figure supplement 1. The image data used in all of these experiments include much larger regions of cells than can be reasonably visualized in a figure panel, and each dataset is composed of multiple experiments. Image panels shown were chosen essentially at random and were cropped to a scale that would allow the viewer to assess the level of cellular detail present in the dataset, so the apparent density in these cases is arbitrary. We did not observe systematic differences in cell density across treatments, and we have now indicated this in the respective figure legend. We have revised the former Figure S1 (now Figure S6) to include larger fields of cells with more similar densities across conditions, but please note that these images still represent only a small fraction of the data (~78,000 cells across all conditions), all of which are represented in the summary plots and model and are included in the supplemental data file for Figure 6. Regarding the trends visible in the former Figure 2B,D (Now Figure 1B,D), we agree that the visible trend is subtle, but it is consistent with the trends shown in the quantification plots, which best represent the full dataset.

4) In some instances, the information provided in the methodology section lacked sufficient detail.

We have reviewed and revised the methodology to ensure complete detail. We have included a description of and reference regarding the sequential staining technique employed to collect many immunofluorescent measurements in the same cells. We also include the customized media formulations in the materials tables. The base medium for all starvation conditions was lacking both glutamine and glucose; where noted (“noAA” formulation) the base medium was also lacking all amino acids.

5) The authors should acknowledge and/or experimentally address limitations of their study pertinent to relying on a single cell line and excluding physiological variables that are known to affect mTORC1 activity (e.g., oxygen levels).

We have included text specifically discussing the limitations that must be considered when interpreting our experiments. However, to mitigate the limited interpretability of data from a single cell line, we have also added experiments employing mouse embryonic fibroblast (MEF) cells, a very different context from the MCF10As used previously. These experiments are presented in Figure 4G.

Reviewer #1 (Recommendations for the authors):– The authors propose that the prevalent model of mTORC1 regulation is that analog to AND gate type of digital logic gate, whereby it is assumed mTORC1 is only active in the presence of both amino acids and growth factors under optimal cellular energy conditions. This is likely to be my subjective interpretation, but I do not think that such AND gate model of mTORC1 regulation is strongly supported by the literature. In contrary, I think that the prevailing model of mTORC1 regulation is that of a rheostat that is akin to the one proposed by the authors. With this in mind, my suggestion to the authors is not to claim the novelty of the model of mTORC1 regulation, but rather their ability to support the existing model by quantifying the effects of various stimuli on mTORC1 activity at the single-cell level, which at least I found to be quite exciting and warranted.

We agree with this perspective and have shifted our focus accordingly; please see the response in the essential revisions section for more details.

– The major limitation of the study is that it based on a single cell line. This renders generality of the authors' conclusions somewhat questionable and should be either accompanied by testing other cell lines or commented as a limitation of the study in the manuscript. Along the same lines, the authors should consider toning down claims of how relative their findings are physiologically as although they study the effects of glucose, amino acids, and insulin on MCF10A cells, it is unlikely that their model reflects the complexity of the physiological environments which comprise fluctuations in oxygen and other factors.

We agree and have added these limitations as the last section in the Discussion, as well as added data from another cell line (MEFs). Please see the response in the essential revisions section for more details.

– Methods section regarding the exact conditions of starvation/stimulation should perhaps be further clarified. In addition, the authors should consider providing evidence that 6h starvation of MCF10A cells is sufficient to shut down PI3K/AKT signaling. Would mTORC1 be activated by amino acids in the presence of e.g., AKT inhibitor? It was also thought that it would be important to monitor potential effects of manipulations in nutrient and/or insulin on reporter expression.

Data addressing this point can be found in Figure 3B and Figure 3 supplement 1 – C. Please see our response to this comment under point 2 of the essential revisions section for more details.

– Phosphorylation of S235/236 on rpS6 can be independent of the mTORC1/S6K axis and its regulation is unaltered in cells lacking S6K1 and 2 (PMID: 15060135). Ergo, using p-S235/236 on rpS6 as a readout of mTORC1 activity may be problematic.

We agree that there is redundancy with other kinases in the regulation of this site, and we have moderated our discussion of this readout to account for this.

Reviewer #2 (Recommendations for the authors):1. Does the amount of nutrients/growth signals added affect total expression of TFEB fusion proteins and mutants? Is this taken into consideration in the calculation of C/N particularly during prolonged periods?

We show a plot of the total amount of TFEB reporter (TFEB C+N, in comparison to TFEB C/N) in Figure S1F-G. In general, changes in reporter expression are on a much more gradual time scale (6 or more hours) and do not have a significant impact on the ratio of cytosolic to nuclear intensity (C/N), which as a proportion inherently adjusts for the total amount of reporter as long as there are no saturation effects. We also observe no evidence that the reporter is saturating within any of the conditions we have tested.

2. The findings in Figures 4 and 5 that TFEB-C/N responds to amino acids despite absence of insulin should be verified for the presence of insulin/growth factor signaling. According to the Methods, cells were starved for 6 hr. There could be residual insulin/PI3K signaling, thus allowing TFEB response upon amino acid addition. They should examine PI3K/Akt signaling. Would this aa response still occur upon inhibition of PI3K or Akt? What happens when starvation is prolonged to 12-16 hr before re-addition of nutrients?

Data addressing this point can be found in Figure 3B and Figure 3 supplement 1 – C. Please see our response to this comment under point 2 of the essential revisions section for more details.

Reviewer #3 (Recommendations for the authors):This is an interesting manuscript from Sparta and colleagues that investigates dynamics of MTOR and TFEB signalling.While this reviewer appreciates the elegance and power of the experimental system, and the substantial amount of data in the manuscript, we did not find that the data provide enough clear insight into the mechanisms of MTOR regulation or signalling to justify publication in eLife. For publication as a full-length paper, it would seem the narrative would need to be re-structured to highlight how findings illustrate novel insights with relevance that can be appreciated by a broad audience. The paper should be revised to focus on a smaller number of specific biological questions that could be addressed by the single cell signalling data.For example, the authors were able to demonstrate that the truncated TFEB-TR reporter behaves similarly to full length, but that a 3xSA-TFEB-TR lacking GSK3beta modification sites was poorly responsive, but these are not significant enough advances for the field.

We thank the reviewer for their careful assessment of the paper and the recognition of the paper’s strengths. We agree with this criticism of the original manuscript, where we did not fully articulate the new insights for a general signal transduction audience. We have now restructured the manuscript and rewritten the Introduction section to better emphasize what quantitative data were lacking in the mTOR field and how these data would be necessary to understand mTOR’s function as a regulator of homeostasis. We have also performed a substantial in-depth analysis of our multiplexed dataset (previously Figure 1, now Figure 6) in which we ask how mTORC1 activity relate to protein synthesis. In this analysis, we show that while mTORC1 readouts are sufficient to almost completely explain protein synthesis activity on average, these readouts only partially predict protein synthesis rate within individual cells, implying the existence of additional regulators that correlate with mTORC1 on the bulk level but not within single cells. We also show that normalized mTORC1 targets are poor individual predictors of protein synthesis within single cells but are highly synergistic, implying that control of protein synthesis by mTORC1 is achieved by regulation of a network of factors rather than any individual downstream effector. Both of these insights are important for fully understanding how mTORC1 acts as a homeostatic regulator of cellular anabolic activity, and we think they will be essential for future efforts to quantitatively model cellular homeostasis.